# Uncleaved prefusion-optimized gp140 trimers derived from analysis of HIV-1 envelope metastability

Leopold Kong[1,2,3,*], Linling He[4,*], Natalia de Val[1,2,3,*], Nemil Vora[4], Charles D. Morris[4], Parisa Azadnia[4], Devin Sok[2,3,4], Bin Zhou[5], Dennis R. Burton[2,3,4,6], Andrew B. Ward[1,2,3], Ian A. Wilson[1,2,3,7,8] & Jiang Zhu[1,3,4]

The trimeric HIV-1 envelope glycoprotein (Env) is critical for host immune recognition and neutralization. Despite advances in trimer design, the roots of Env trimer metastability remain elusive. Here we investigate the contribution of two Env regions to metastability. First, we computationally redesign a largely disordered bend in heptad region 1 (HR1) of SOSIP trimers that connects the long, central HR1 helix to the fusion peptide, substantially improving the yield of soluble, well-folded trimers. Structural and antigenic analyses of two distinct HR1 redesigns confirm that redesigned Env closely mimics the native, prefusion trimer with a more stable gp41. Next, we replace the cleavage site between gp120 and gp41 with various linkers in the context of an HR1 redesign. Electron microscopy reveals a potential fusion intermediate state for uncleaved trimers containing short but not long linkers. Together, these results outline a general approach for stabilization of Env trimers from diverse HIV-1 strains.

[1] Department of Integrative Structural and Computational Biology, The Scripps Research Institute, La Jolla, California 92037, USA. [2] International AIDS Vaccine Initiative Neutralizing Antibody Center and the Collaboration for AIDS Vaccine Discovery, The Scripps Research Institute, La Jolla, California 92037, USA. [3] Scripps Center for HIV/AIDS Vaccine Immunology & Immunogen Discovery, The Scripps Research Institute, La Jolla, California 92037, USA. [4] Department of Immunology and Microbial Science, The Scripps Research Institute, La Jolla, California 92037, USA. [5] Department of Chemistry, The Scripps Research Institute, La Jolla, California 92037, USA. [6] Ragon Institute of Massachusetts General Hospital, Massachusetts Institute of Technology and Harvard, Cambridge, Massachusetts 02139-3583, USA. [7] The Joint Center for Structural Genomics, The Scripps Research Institute, La Jolla, California 92037, USA. [8] Skaggs Institute for Chemical Biology, The Scripps Research Institute, La Jolla, California 92037, USA. * These authors contributed equally to this work. Correspondence and requests for materials should be addressed to I.A.W. (email: wilson@scripps.edu) or to J.Z. (email: jiang@scripps.edu).

A major goal of vaccine development for human immunodeficiency virus type-1 (HIV-1) is to induce broadly neutralizing antibodies (bNAbs) by vaccination[1]. All bNAbs identified thus far target the envelope glycoprotein (Env) trimer on the surface of HIV-1 virions[2]. The precursor Env protein, gp160, is trafficked from the endoplasmic reticulum to the Golgi and cleaved by cellular proteases of the furin family into its mature form[3]. The cleaved Env trimer engages host receptors to mediate viral entry and is the primary target of protective humoral immune responses. The functional Env is a trimer of heterodimers, each containing a receptor-binding protein, gp120, and a transmembrane fusion protein, gp41, which are held together by non-covalent interactions[4]. The mature Env is also metastable, as it is poised to undergo dramatic and irreversible conformational changes upon receptor and co-receptor binding to mediate membrane fusion. This inherent metastability facilitates immune evasion by inducing gp120 shedding[5] and generating a diverse assortment of native, more open and non-native Env conformations[6].

Various strategies have been proposed to overcome Env metastability and to create stable, homogeneous gp140 trimers for structural and vaccine studies. One approach involved deletion of the cleavage site between gp120 and the gp41 ectodomain (gp41$_{ECTO}$), and addition of trimerization motifs to the C terminus of gp41$_{ECTO}$ to stabilize the trimer[7–9]. Another approach covalently linked the cleaved gp120 and gp41$_{ECTO}$ domains with an engineered disulfide bond (termed SOS), destabilized the gp41 postfusion conformation through addition of an I559P mutation (termed IP) and improved trimer solubility by truncating the hydrophobic membrane-proximal external region at residue 664 (ref. 10). This latter design, designated SOSIP.664, when applied to clade-A BG505 (ref. 11), produced a stable, soluble and cleaved gp140 trimer with an outstanding antigenic profile[12,13] and excellent structural mimicry of the native spike[14–16]. The atomic structures of BG505 SOSIP.664 trimer from X-ray crystallography[14,16] and cryo-electron microscopy (EM)[15] provided a detailed picture of this long-sought vaccine target. Some bNAbs that were previously crystallized with gp120 core and scaffolded V1V2, such as PGV04 and PG9, have now been found to interact with other structural elements present only on the native trimer to enhance recognition of native virions[14,15,17]. Using the SOSIP trimer as a sorting probe, new bNAbs have been identified and characterized[18–22]. The SOSIP design has also been extended to other HIV-1 strains[23–25] and permitted the incorporation of additional stabilizing mutations[26,27]. Recently, the immunogenicity of SOSIP trimers in rabbits and non-human primates was reported, paving the way for human vaccine trials[28]. While the full benefit of retaining cleavage between gp120 and gp41 has been demonstrated for the SOSIP trimer, the utility of flexible linkers at the cleavage site has also been successfully explored with the design of single-chain gp140 (sc-gp140)[29] and native flexibly linked (NFL) trimers[30,31].

Furin co-expression and bNAb affinity purification are required for the production of soluble, cleaved SOSIP trimers[11]. Negative selection[29,30] and multi-cycle size exclusion chromatography (SEC)[29] have recently been reported for purification of well-folded, uncleaved gp140 trimers. The complexity in production of native-like trimers has limited the use of nucleic acid vaccine platforms[32–34] and also raised the question of whether we have adequately addressed the causes of HIV-1 Env metastability. In this study, we investigate the primary causes of HIV-1 Env metastability and explore alternative trimer designs. We hypothesize that the disorder observed at the HR1 N terminus (residues 547–569) is indicative of metastability that could potentially be minimized by protein engineering. To this end, we evaluate 10 BG505 gp140 trimers with computationally redesigned HR1. These constructs show substantially higher trimer yield and purity, with native SOSIP-like properties demonstrated by crystal structures, negative-stain EM and antibody binding. We next examine the structural and antigenic effects of replacing the furin cleavage site between gp120 and gp41 with a linker in the context of a selected HR1 redesign. Our analyses uncover the sensitivity of gp140 folding to modification of this proteolytic site, with a potential fusion intermediate state observed for trimers with short linkers lacking the SOS mutation. By contrast, the HR1-redesigned trimers with a long linker, termed uncleaved prefusion-optimized (UFO) trimers, adopt a native-like conformation that retains many salient features of the SOSIP trimer. Finally, we demonstrate the utility of a generic HR1 linker in trimer stabilization for diverse HIV-1 strains. Our rational approach towards redesign of two critical regions of gp140—HR1 and the cleavage site—provides valuable insights into HIV-1 Env metastability as well as novel platforms for trimer-based vaccine design.

## Results

**Ensemble-based design of the HR1 N terminus**. We hypothesized that the N terminus of HR1 (residues 547–569) is a critical determinant of HIV-1 trimer metastability because it is poised to elongate the HR1 helix during fusion and is disordered in all but one crystal structure of the SOSIP trimer[35] and one cryo-EM structure of full-length, wild-type (WT) Env[36], where it still appears less ordered in comparison with the surrounding regions (Fig. 1a). Such disorder at the top of the long HR1 central helix may seem unexpected since this region resides in the core of the Env complex; however, because this region is expected to form a helix in the postfusion state, as in the equivalent region of influenza hemagglutinin and other type 1 viral fusion proteins[37], it must remain less ordered or adopt a different conformation in the prefusion state. For the SOSIP design, in addition to an engineered disulfide bond (A501C/T605C), the I559P mutation was introduced to destabilize the postfusion state[10] and was critical for the production of high-quality trimers, strongly supporting the notion that this HR1 region might be related to metastability. In this study, the HR1 bend was rationally redesigned to stabilize the prefusion conformation rather than to destabilize the postfusion conformation as in the SOSIP trimer (Fig. 1b). Although this WT HR1 region consists of 21 residues, the Cα distance between G547 and T569 is merely 24.8 Å, which is equivalent to a fully extended polypeptide backbone of only 6.3 residues. Here we examined two loop lengths—8 and 10 residues—for the HR1 redesign, allowing for a small degree of flexibility while markedly shortening the WT HR1 loop. We utilized ensemble-based *de novo* protein design (see Methods) to identify sequences that may stabilize the prefusion conformation (Fig. 1c). Given a specified loop length, a large ensemble of backbone conformations was generated to bridge the gap between G547 and T569 (Supplementary Fig. 1a). For 8-residue loops, the Cα root-mean-square fluctuation (RMSF) ranged from 1.3 to 5.7 Å with an average of 2.3 Å, whereas 10-residue loops exhibited an average Cα RMSF of 3.6 Å (Supplementary Fig. 1b). After an exhaustive sampling in sequence space, all designs were ranked by their energy scores (Supplementary Fig. 1c). The 5 top-ranking sequences for each loop length, totaling 10, were advanced to experimental validation (Supplementary Fig. 1d).

**Biochemical and biophysical analyses of HR1 redesigns**. As demonstrated for SOSIP, sc-gp140 and NFL trimers, biochemical and biophysical properties can provide an initial assessment of

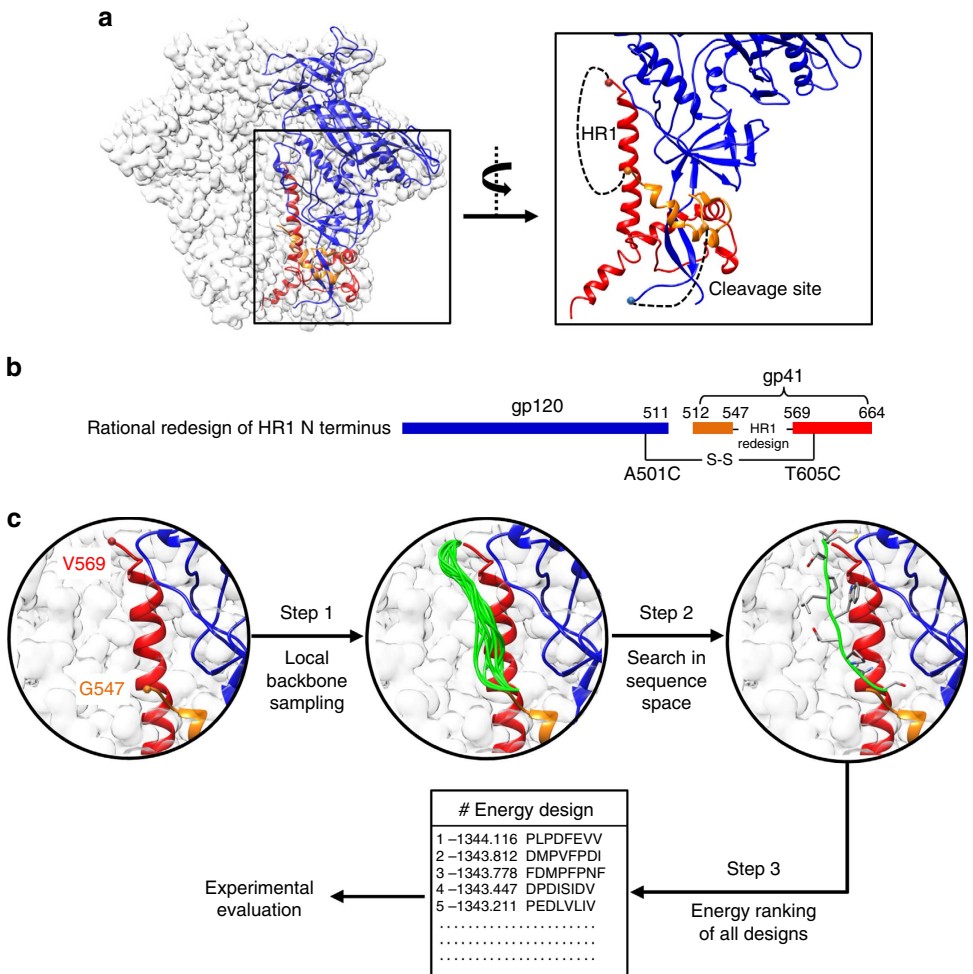

**Figure 1 | Computational redesign of the HR1 N terminus and cleavage site in the context of a prefusion Env structure. (a)** Atomic model and molecular surface of WT BG505 SOSIP.664 trimer (PDB ID: 4TVP) with gp120 and two regions of gp41$_{ECTO}$ (residues 518–547 and 569–664) within one gp140 protomer coloured in blue, orange and red, respectively. A zoomed-in view of the gp140 structure surrounding the disordered HR1 N terminus (residues 547–569) and the cleavage site-containing region (residues 505–518) is shown on the right with the two structural gaps connected by black dotted lines. **(b)** Schematic representation of HR1 redesign. **(c)** Computational procedure used for ensemble-based *de novo* protein design of the HR1 region (residues 547–569). After a local backbone sampling in torsion space (step 1) and an exhaustive search in sequence space (step 2), redesigned HR1 sequences are ranked by energy (step 3) before the manual selection of candidates for experimental validation.

trimer designs[11,29,30]. Following a similar strategy, we assessed the 10 HR1-redesigned BG505 trimers containing the same T332N (to restore the N332 epitope), SOS (A501C/T605C) and R6 mutations as in the SOSIP.664 trimer (except for I559P). As noted, different purification protocols can produce trimers of varying quality. Here we adopted a rather simple protocol utilizing materials that are readily available to most researchers. All constructs were expressed transiently in HEK293F cells with co-transfected furin as previously described[11]. The secreted Env proteins were purified using a *Galanthus nivalis* lectin (GNL) column followed by SEC on a Superdex 200 10/300 column. One-litre expression produced sufficient quantities (3–7 mg) of HR1-redesigned trimers, compared with three 2-l expressions for the SOSIP trimer. Although GNL purification does not yield the purest trimers[24], it enables comparison of basic properties between various trimer constructs such as monomer, dimer and higher multimeric species (aggregates) that would otherwise be filtered out by more sophisticated purification methods.

We compared the SEC profiles through simple metrics, utilizing ultraviolet (UV) 280 nm absorbance values (Fig. 2a). The UV value of the trimer peak was used as an indicator of the

trimer yield, with the aggregate and dimer/monomer peaks measured as percentages of their UV values versus that of the trimer peak. The 2-l SOSIP expression showed an average UV value of 371 for the trimer peak, with average percentages of 31% and 49% for the aggregate and dimer/monomer peaks, respectively (Fig. 2a, top). The five 8-residue HR1 redesigns showed significantly increased trimer yield with reduced aggregate and dimer/monomer peaks in the SEC profiles (Fig. 2a, middle). Overall, HR1 redesigns 1 and 2 were the best performers in this group. For example, the HR1 redesign 2 showed a near-twofold increase in the trimer peak, with a 16 and 22% reduction for aggregate and dimer/monomer peaks relative to SOSIP, indicative of improvement in both trimer yield and purity. The five 10-residue HR1 redesigns showed a similar but less pronounced improvement (Fig. 2a, bottom). Notwithstanding, HR1 redesign 10 showed a UV value for the trimer peak that is comparable to the SOSIP trimer from 2-l expression, with the same low level of unwanted Env species as HR1 redesigns 1 and 2. This finding was consistent with the blue native polyacrylamide gel electrophoresis (BN-PAGE) that showed more intense trimer bands on the gel (Supplementary Fig. 2a). The trimer-containing fractions eluted at 10.25–10.75 ml were used for the initial assessment of trimer

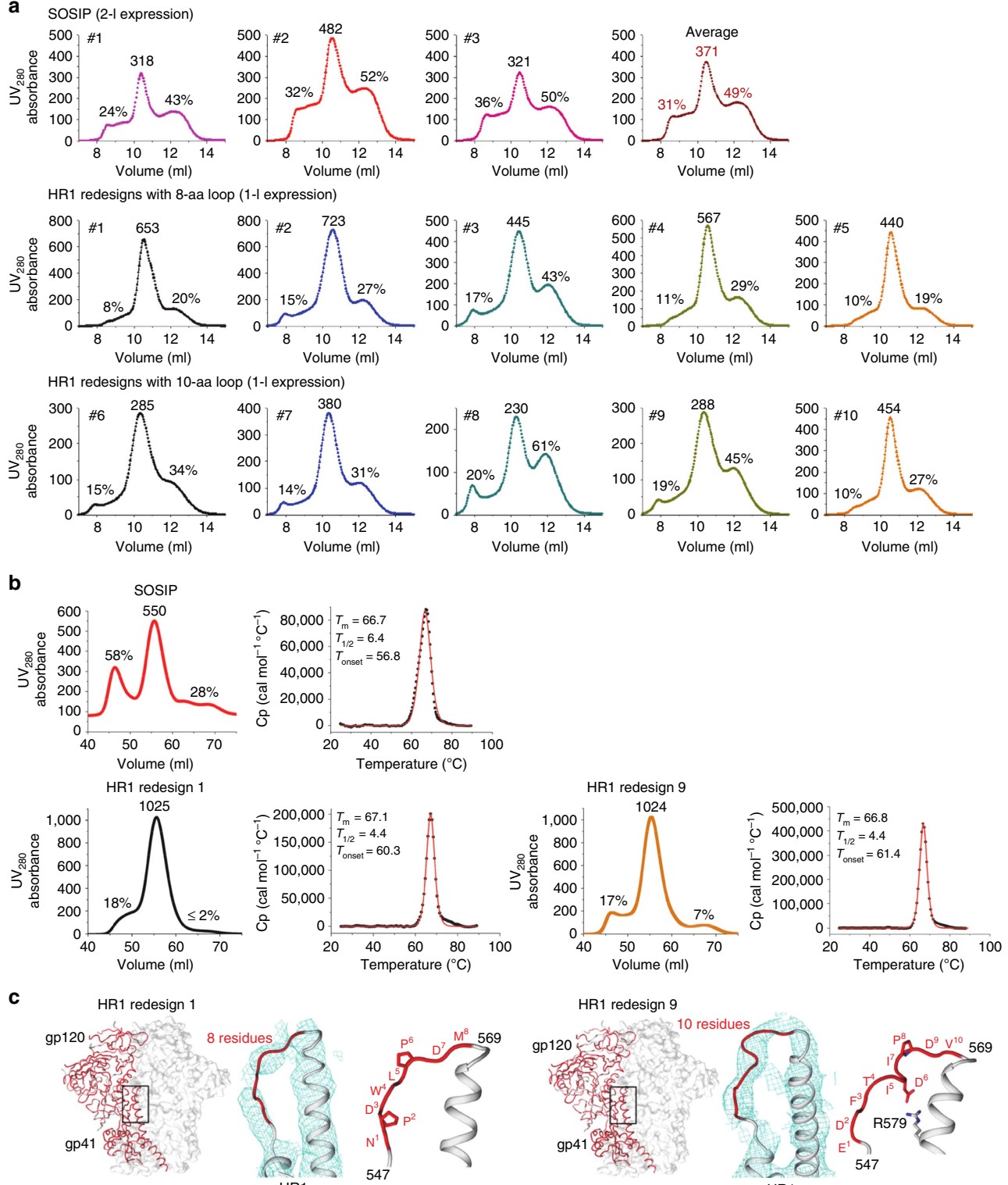

**Figure 2 | Biophysical properties and crystal structures of HR1-redesigned Env trimers.** (**a**) SEC profiles of 293F-expressed, GNL-purified WT BG505 SOSIP.664 trimer (top), five 8-residue HR1 redesigns (middle) and five 10-residue HR1 redesigns (bottom) from a Superdex 200 10/300 column. Two- and one-litre transient expressions were used for the SOSIP trimer and all HR1 redesigns, respectively. The absolute $UV_{280}$ absorbance value of the trimer peak and the percentages of UV values for aggregate peak (at 9 ml) and dimer/monomer peak (at 12 ml) relative to the trimer peak (at 10.5 ml) are labelled for comparison. (**b**) SEC and DSC profiles of 293S-expressed, 2G12-purified SOSIP trimer (upper panel) and two selected HR1 redesigns (1 and 9, lower panel). The absolute $UV_{280}$ absorbance value of the trimer peak and the percentages of UV values for aggregate peak (at 50 ml) and dimer/monomer peak (at 62.5 ml) relative to the trimer peak (at 55 ml) are labelled on the SEC profiles from a Superdex 200 16/600 column. Three thermal parameters ($T_m$, $T_{1/2}$ and $T_{onset}$) are labelled on the trimer DSC profiles. (**c**) Crystal structures of HR1-redesigned trimers (1 and 9) determined as complexes with Fabs PGT128 and 8ANC195. For both trimers, an overall view of the structure is shown on the left with a zoomed-in view of the backbone for the redesigned HR1 loop on the right. The structures of the entire complexes are shown in Supplementary Fig. 3c.

stability by differential scanning calorimetry (DSC). For all 10 tested HR1 redesigns, we observed similar unfolding peaks with a thermal denaturation midpoint ($T_m$) ranging from 65.7 to 69.2 °C (Supplementary Fig. 2b), closely resembling the $T_m$ of 68.1 °C reported for the SOSIP trimer[11].

Overall, redesign of this shortened HR1 region exerted a positive effect on the Env trimers. While the HR1 redesigns retained the thermal stability of parent SOSIP.664 trimer, they also increased the trimer yield and reduced other undesirable Env species, supporting the notion that this HR1 connecting loop is a key determinant of HIV-1 trimer metastability.

**Crystallographic analysis of two HR1 redesigns.** HR1 redesigns 1 and 9 were selected for crystallographic analysis. These two constructs differed in their redesigned loop lengths (8 versus 10 residues) and SEC profiles, with a higher dimer/monomer peak observed for HR1 redesign 9 (Fig. 2a), which provided an opportunity to examine how HR1 loop length and design variation affect trimer structure. Owing to the stringent requirement of sample homogeneity for crystallization, we prepared the HR1-redesigned and WT SOSIP trimers as previously described[38]. In brief, all trimers were produced in N-acetylglucosaminyltransferase I-negative (GnTI$^{-/-}$) HEK293S cells and purified using a 2G12 affinity column followed by SEC on a Superdex 200 16/600 column. For the WT SOSIP trimer, the SEC profile displayed a notable aggregate peak of high molecular weight with a UV value that is 58% of the trimer peak, and a lower peak containing gp140 monomers and dimers (Fig. 2b, upper panel). By contrast, the 2G12-purified HR1 redesigns showed a marked improvement in trimer yield and purity (Fig. 2b, lower panel). Of note, HR1 redesign 1 showed an almost undetectable level of monomers and dimers, whereas HR1 redesign 9 still contained a small fraction of impurities. Nevertheless, the SEC profiles of 293S-expressed, 2G12-purified trimers are consistent with those of the 293F-expressed, GNL-purified trimers described above (Fig. 2a). This finding was further confirmed by BN-PAGE of SEC fractions (Supplementary Fig. 3a,b) and thermal stability of the modified trimers measured by DSC (Fig. 2b), suggesting that the improved trimer properties are an intrinsic feature of the HR1 redesigns and independent of the expression and purification systems.

Co-crystallization with antigen-binding fragments (Fabs) of PGT128 and 8ANC195 yielded Env complex structures at resolutions of 6.9 and 6.3 Å for HR1 redesigns 1 and 9, respectively (Table 1 and Supplementary Fig. 3c,d). Overall, the redesigned trimers showed nearly identical structures to that of the SOSIP trimer[14–16] at this modest resolution, with Cα root-mean-square deviations (RMSD) <0.25 Å (Fig. 2c). Thus, the results confirmed that gp140 trimers with shortened and redesigned HR1 still adopt a native-like prefusion conformation. The limited resolution of the crystal structure allowed us to only determine the approximate backbone conformation of the redesigned HR1 loop, but nevertheless provided insights into how these two distinct designs stabilize the prefusion trimer. We speculate that the shortened loop (8 or 10 versus 21 residues in WT) and redesigned sequence disrupt the heptad repeat motif that extends the HR1 helix in the postfusion state, thereby stabilizing the prefusion form. Furthermore, both HR1 redesigns contained proline, at positions 2 and 6 in the 8-residue loop and at position 8 in the 10-residue loop (Fig. 2c), which likely increases the rigidity of the backbone. Of note, Asp 6 in HR1 redesign 9 is poised to form a salt bridge with Arg 579 of the neighbouring HR1 helix, stabilizing the slightly turned loop (Fig. 2c, right). In conclusion, gp140 appears to be highly tolerant

of the HR1 redesign, which can enhance the efficiency of protein production without sacrificing overall structural integrity.

**Antigenic profiling of HR1-redesigned BG505 gp140 trimers.** The BG505 SOSIP.664 trimer represents a close mimic of the native spike for immune recognition by antibodies[11–13]. Here we sought to investigate whether the HR1 redesign would affect Env trimer binding to bNAbs and non-NAbs using bio-layer interferometry (BLI) with immunoglobulin G (IgG). Again, we studied trimers prepared using a simple GNL purification so that we could compare the basic properties of different trimer constructs. BN-PAGE of SEC fractions obtained from a Superdex 200 16/600 column following GNL purification was performed to facilitate the selection of well-folded trimers for antigenic profiling (Fig. 3a). In this context, we also characterized the HR1 redesign 1 by negative-stain EM. In the unliganded state, a reconstruction at ~22-Å resolution displayed a morphology closely resembling that of the SOSIP trimer prepared using the same protocol (Fig. 3b and Supplementary Fig. 4). The agreement of crystal and EM structures further confirmed the integrity of HR1-redesigned trimers before antigenic characterization.

First, we measured trimer binding to a panel of representative bNAbs (Fig. 3c and Supplementary Fig. 5a). We utilized V1V2 apex-directed, quaternary bNAbs PGDM1400 (ref. 22), PGT145 (ref. 39) and PG16 (ref. 40) to examine whether the trimeric structure with associated glycan shield was native-like. For PGDM1400, HR1 redesigns 1 and 9 displayed faster on- and off-rates than WT SOSIP, with $K_D$ ($k_{off}/k_{on}$) values ranging from 7 to 11 nM. A similar pattern was observed for PG16 and PGT145. For VRC01, a representative of a class of CD4-binding site (CD4bs)-directed bNAbs[41–44], all three trimers showed nearly identical binding profiles, suggesting that HR1 redesign had little effect on the presentation of this conserved site of vulnerability. A similar pattern was also seen for NAb b12, which engages the CD4bs with a different angle of approach than VRC01 (ref. 45). For bNAbs targeting the V3 stem and surrounding glycans, including N332 (PGT121, PGT128 and PGT135 (ref. 39)), and the high-mannose gp120 glycan cluster (2G12 (ref. 46)), all three trimers showed identical binding profiles, indicating that these glycan epitopes remained intact upon HR1 redesign. Finally, we measured trimer binding to two bNAbs that recognize conformational epitopes spanning regions in both gp120 and gp41 (refs 19,21). All three trimers bound strongly to PGT151 with a fast on-rate and a flat dissociation curve, while subtle differences were observed in 35O22-binding kinetics.

Next, we measured trimer binding to a panel of representative non-NAbs (Fig. 3d and Supplementary Fig. 5b). All three tested trimers bound to CD4bs-specific b6 and F105. The HR1-redesigned trimers displayed weaker binding to F105 than did the SOSIP trimer, with a slightly faster off-rate detected for HR1 redesign 1. However, no differences in kinetics were observed for b6. For V3-specific 19b and 447-52D, all three trimers showed fast association and slow dissociation, indicative of some V3 exposure in the trimer preparation. Previously, 19b was found to bind the SOSIP trimer by enzyme-linked immunosorbent assay (ELISA), but only to a limited extent by EM and surface plasmon resonance (SPR)[11,12]. In this study, SPR analysis of 2G12-purified BG505 SOSIP.664 trimer showed consistent results (Supplementary Fig. 6). The differences in experimental set-ups such as analyte immobilization (see Methods and ref. 12) may have contributed to the more visible V3 exposure by BLI. Nevertheless, the V3 exposure may be minimized by conformational fixation as recently demonstrated for the SOSIP trimer[26,27]. We then tested F240 and 7B2, which recognize the immunodominant epitopes in cluster I of

**Table 1 | X-ray crystallographic data collection and refinement statistics.**

| Data collection | HR1 redesign 1 + Fabs 8ANC195 and PGT128* | HR1 redesign 9 + Fabs 8ANC195 and PGT128* |
|---|---|---|
| X-ray source | SSRL 12-2 | APS 23ID-D |
| Wavelength (Å) | 0.980 | 1.033 |
| Space group | I23 | I23 |
| Unit cell parameters (Å) | $a=b=c=262.0$ | $a=b=c=266.3$ |
| Resolution (Å) | 50.0–6.90 (7.15–6.90)[†] | 50.0–6.30 (6.52–6.30)[†] |
| Observations | 96,139 | 104,666 |
| Unique reflections | 5,022 (496)[†] | 6,914 (685)[†] |
| Redundancy | 19.1 (20.3)[†] | 15.1 (15.7)[†] |
| Completeness (%) | 100.0 (100.0)[†] | 100.0 (100.0)[†] |
| $\langle I/\sigma_I\rangle$[‡] | 15.5 (1.3)[†] | 17.9 (2.3)[†] |
| $R_{sym}$[§] | 0.16 (4.21)[†] | 0.10 (2.09)[†] |
| $R_{pim}$[§] | 0.05 (0.81)[†] | 0.04 (0.54)[†] |
| $CC_{1/2}$[§] | 0.83 (0.33)[†] | 0.88 (0.51)[†] |
| | | |
| *Refinement statistics* | | |
| Resolution (Å) | 47.83–6.92 (7.61–6.92)[†] | 40.14–6.31 (6.79–6.31)[†] |
| Reflections (work) | 4,519 (1,135)[†] | 6,208 (1,236)[†] |
| Reflections (test) | 492 (109)[†] | 684 (139)[†] |
| $R_{cryst}$ (%)[||] | 28.4 | 28.1 |
| $R_{free}$ (%)[¶] | 32.2 | 32.2 |
| Average $B$ value (Å²) | 292 | 350 |
| Wilson $B$ value (Å²) | 407 | 356 |
| Coordinate error (Å)[#] | 1.3 | 1.1 |
| | | |
| *R.m.s.d. from ideal geometry* | | |
| Bond length (Å) | 0.004 | 0.004 |
| Bond angles (°) | 0.88 | 0.84 |
| | | |
| *Ramachandran statistics (%)*[**] | | |
| Favoured | 95.2 | 95.1 |
| Outliers | 0.1 | 0.2 |
| PDB ID | 5JS9 | 5JSA |

R.m.s.d., root-mean-square deviation.
*One crystal was used for each data set.
†Numbers in parentheses refer to the highest-resolution shell.
‡Calculated as average (*I*)/average (σ*I*).
§$R_{sym}=\Sigma_{hkl}\Sigma_i|I_{hkl,i}-\langle I_{hkl}\rangle|/\Sigma_{hkl}\Sigma_i I_{hkl,i}$, where $I_{hkl,i}$ is the scaled intensity of the *i*th measurement of reflection *h, k* and *l*; $\langle I_{hkl}\rangle$ is the average intensity for that reflection; and *n* is the redundancy. $R_{pim}$ is a redundancy-independent measure of the quality of intensity measurements. $R_{pim}=\Sigma_{hkl}(1/(n-1))^{1/2}\Sigma_i|I_{hkl,i}-\langle I_{hkl}\rangle|/\Sigma_{hkl}\Sigma_i I_{hkl,i}$, where $I_{hkl,i}$ is the scaled intensity of the *i*th measurement of reflection *h, k* and *l*; $\langle I_{hkl}\rangle$ is the average intensity for that reflection; and *n* is the redundancy. $CC_{1/2}=$ Pearson Correlation Coefficient between two random half datasets.
||$R_{cryst}=\Sigma_{hkl}|F_o-F_c|/\Sigma_{hkl}|F_o|\times100$.
¶$R_{free}$ was calculated as for $R_{cryst}$, but on a test set comprising 10% of the data excluded from refinement.
#Calculated by the Phenix refinement package using a maximum likelihood based method (ref. 58).
**Values calculated using MolProbity (http://molprobity.biochem.duke.edu/).

gp41$_{ECTO}$. The SOSIP trimer appeared to bind both antibodies at a low level with a slight preference for F240. Interestingly, the two HR1 redesigns showed reduced binding to F240 and almost negligible binding to 7B2, indicating a more closed or less flexible gp41$_{ECTO}$. We also investigated two CD4i antibodies, 17b and A32. All three trimers showed no binding to 17b in the absence of soluble CD4 (sCD4), with HR1 redesign 1 exhibiting only a minimal level of A32 recognition although all trimers bound very weakly to this antibody.

Overall, the two HR1 redesigns displayed broadly similar patterns in their recognition by bNAbs with the exception of altered kinetics for apex-directed quaternary bNAbs. While all three trimers showed some V3 exposure, the two HR1 redesigns appeared to shield non-neutralizing gp41$_{ECTO}$ epitopes more effectively than the SOSIP trimer. The observed binding to non-NAbs may be attributed to the use of IgG instead of Fab that can substantially enhance avidity and the use of a different immobilization strategy in the BLI experiment.

**Replacing the furin cleavage site with short linkers.** Sharma et al.[30] reported a native-like, cleavage-independent gp140 trimer designated NFL. In another study, Georgiev et al. replaced the

cleavage site between gp120 and gp41 with linkers of up to 20 residues (termed sc-gp140)[29]. Although the presence of aberrant structures was speculated for sc-gp140 trimers with short linkers[29], the precise effect of cleavage site modification on gp140 folding and structure remained unclear. We addressed this critical issue in the context of HR1 redesign 1, which had been validated both structurally and antigenically (Figs 2 and 3).

We first examined the outcome of replacing the cleavage site-containing region (residues 500–519) with a redesigned connecting loop between gp120 and gp41 (Fig. 4a). The Cα distance between R500 and F519 is 16.8 Å, equivalent to a fully extended backbone of 4.4 residues. Ensemble-based protein design yielded a large pool of 8-residue loops connecting R500 and F519 (Supplementary Fig. 7a). Of note, this design strategy was rather aggressive because these loops may pack differently than the uncleaved WT sequence due to a 10-residue truncation in this region and exclusion of the SOS mutation, as A501 was now part of the region to be redesigned (accordingly, the T605C mutation was reversed). The five top-ranking designs (termed CST1-5, Supplementary Fig. 7b) were characterized by SEC on a Superdex 200 10/300 column following transient expression in HEK293F cells without furin and GNL purification (Fig. 4b,

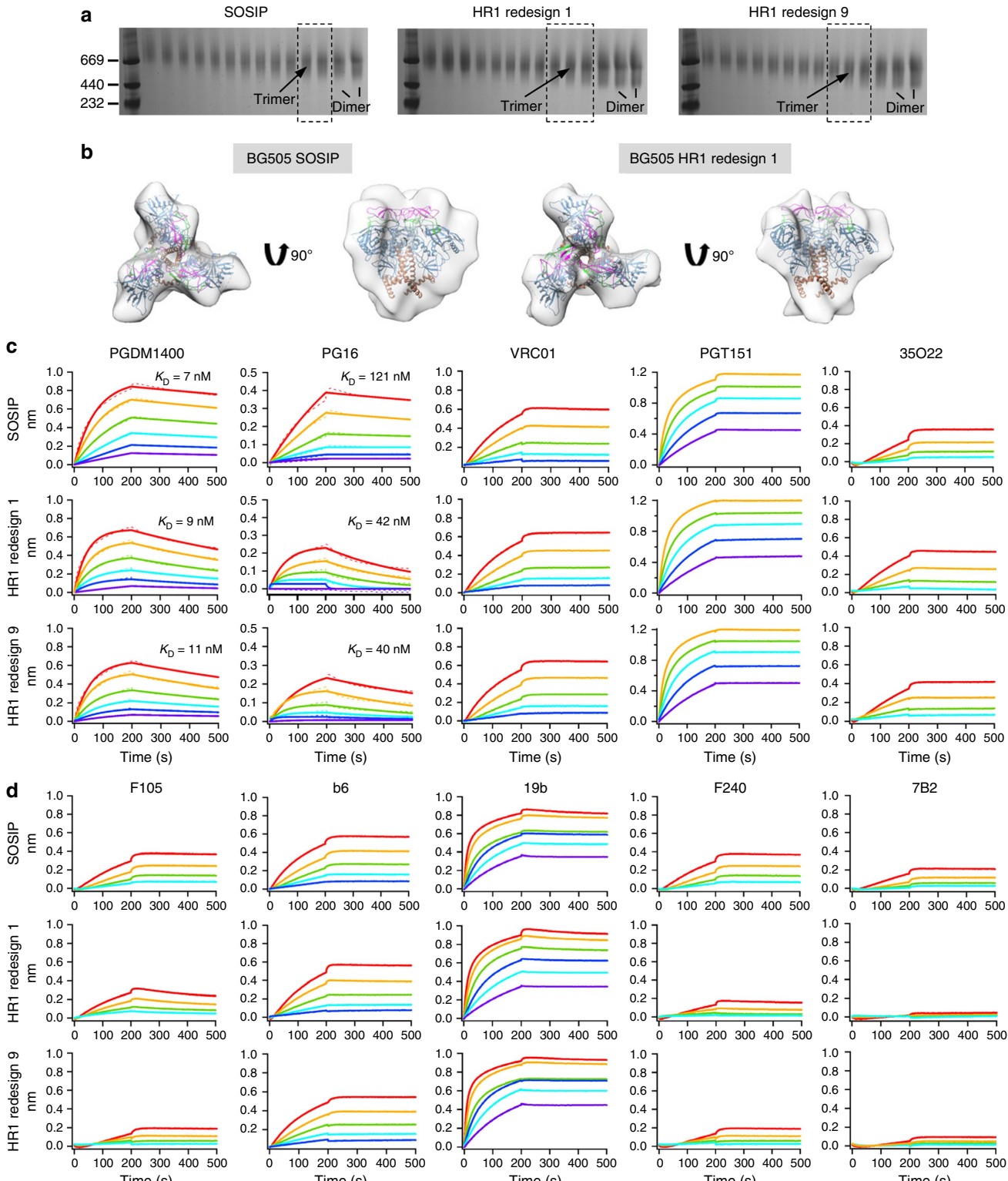

**Figure 3 | EM analysis and antigenic profiling of HR1-redesigned Env trimers.** (**a**) BN-PAGE of 293F-expressed, GNL-purified WT BG505 SOSIP.664 trimer and two HR1 redesigns following SEC separation on a Superdex 200 16/600 column. The fractions used for EM analysis and antigenic profiling are circled by black dotted lines with the expected positions of trimer and dimer bands labelled on the gel. (**b**) 3D reconstructions of 293F-expressed, GNL-purified SOSIP trimer (left) and HR1 redesign 1 (right) derived from negative-stain EM. The trimer densities are shown in grey transparent surface with the fitted high-resolution cryo-EM structure of the SOSIP trimer (PDB 3J5M, gp120 in blue with V1V2 in magenta, V3 in green and gp41 in brown). Both top and side views are shown after fitting the previously published EM model (PDB ID: 3J5M) into the density. Antigenic profiles of the SOSIP trimer and two HR1 redesigns measured for a panel of representative (**c**) bNAbs and (**d**) non-NAbs, with additional antibody binding profiles shown in Supplementary Fig. 5. Sensorgrams were obtained from an Octet RED96 using a trimer titration series of six concentrations (200–12.5 nM by twofold dilution). $K_D$ values calculated from 1:1 global fitting are labelled for V1V2 apex-directed bNAbs (PGDM1400 and PG16) in **c**.

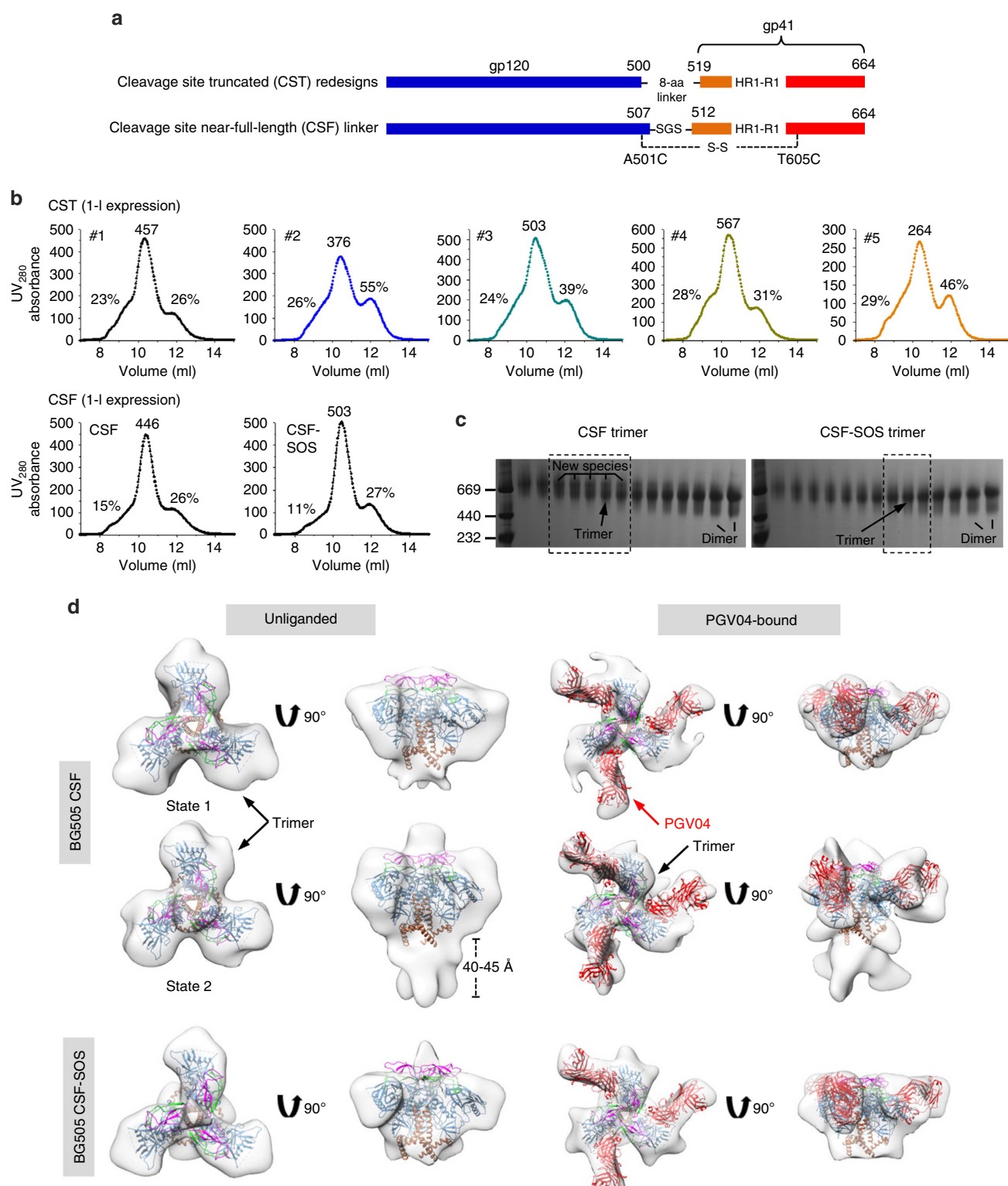

**Figure 4 | Design and characterization of HR1-redesigned Env trimers with short cleavage site linkers.** (**a**) Schematic representation of CST redesigns and near-full-length (CSF) linkers. (**b**) SEC profiles of five CST trimers (obtained from ensemble-based *de novo* protein design, upper panel) and two CSF trimers (lower panel) from a Superdex 200 10/300 column. The UV value of the trimer peak and the percentages of UV values for aggregate peak (at 9 ml) and dimer/monomer peak (at 12 ml) relative to the trimer peak (at 10.5 ml) are labelled for comparison. (**c**) BN-PAGE of 293F-expressed, GNL-purified CSF trimers following SEC separation on a Superdex 200 16/600 column. The fractions used for EM analysis and antigenic profiling are circled by black dotted lines with expected positions for trimer and dimer bands labelled on the gel. New Env species observed for the CSF trimer are also labelled. (**d**) 3D reconstructions of CSF (states 1 and 2) and CSF-SOS trimers in unliganded and PGV04-bound forms derived from negative-stain EM. The trimer densities are shown in grey transparent surface with the fitted high-resolution cryo-EM structure of the SOSIP trimer (PDB 3J5M, gp120 in blue with V1V2 in magenta, V3 in green, gp41 in brown and PGV04 Fab in red). Both top and side views are shown after fitting the previously published EM model (PDB ID: 3J5M) into the density.

upper panel). Overall, CST1-5 showed reduced trimer yield, as well as increased aggregates compared with the parent HR1 redesign 1, as indicated by a higher shoulder left of the main trimer peak in the SEC profiles. For all five cleavage site truncated (CST) constructs, an extra band was observed in BN-PAGE, suggesting the presence of an uncharacterized Env species in the produced proteins (Supplementary Fig. 7c).

We next examined the effect of replacing the cleavage site $_{508}$REKR$_{511}$ with a near-full-length serine-glycine-serine (SGS) linker (termed CSF) (Fig. 4a). Interestingly, CSF displayed a notably reduced aggregate peak in the SEC profile compared with CST1-5, which was further improved by adding back the SOS mutation (termed CSF-SOS; Fig. 4b, lower panel). Similar to CST1-5, an extra band was observed for the CSF trimer in BN-PAGE of trimer-containing fractions after SEC on a Superdex 200 16/600 column (Fig. 4c), suggesting a common pattern associated with short cleavage site linkers. To identify this unknown Env species, we used negative-stain EM to obtain three-dimensional (3D) reconstructions for the CSF trimer (Fig. 4d and Supplementary Fig. 8). Remarkably, two distinct morphologies were observed for the unliganded trimer: one in the prefusion state (at $\sim 20$ Å) similar to the SOSIP trimer and the other in a non-prefusion state ($\sim 17$ Å) that has not been reported previously (Fig. 4d). This non-prefusion trimer conformation (state 2) contains an extended gp41 ($\sim 40-45$ Å) and is referred to as a putative 'fusion intermediate' hereafter. The $\sim 20$ Å EM reconstructions of PGV04-bound CSF trimers in the two different states showed some unoccupied densities that could not be interpreted at this resolution (Fig. 4d). By contrast, a single conformation was observed for the EM reconstruction of the CSF-SOS trimer in both unliganded ($\sim 21$ Å) and PGV04-bound ($\sim 20$ Å) forms. In summary, with short cleavage site linkers the CST and CSF trimers contained a putative fusion intermediate state that could be effectively suppressed by the SOS mutation.

We then examined the antigenicity of CSF and CSF-SOS trimers using a small antibody panel (Supplementary Fig. 9). For bNAbs, we utilized PGDM1400, VRC01 and PGT151, which target the V1V2 apex, CD4bs and gp120-gp41 interface, respectively (Supplementary Fig. 9a). CSF and CSF-SOS bound to PGDM1400 with similar kinetics and affinities, but with a reduced signal observed for CSF, which contains two trimer forms. CSF and CSF-SOS exhibited identical VRC01-binding profiles compared with SOSIP, suggesting that the CD4bs is equally accessible in all of these trimers. For PGT151, CSF and CSF-SOS showed reduced binding with a notable off-rate, suggesting that the linker between gp120 and gp41 may affect PGT151 binding, which is cleavage dependent[18,19]. Three non-NAbs were also tested (Supplementary Fig. 9b). CSF bound more strongly to CD4bs-directed F105 than CSF-SOS due to the presence of putative fusion intermediates. For V3-directed 19b, both CSF trimers displayed similar binding profiles relative to the SOSP trimer and HR1 redesign 1. By contrast, CSF showed enhanced binding to gp41-directed F240 that was effectively reduced by the SOS mutation in CSF-SOS.

**Replacing the furin cleavage site with long linkers**. On the basis of our analysis thus far and the reports on NFL and sc-gp140 trimers[29,30], we hypothesized that combining HR1 redesign with a long cleavage site linker may overcome the tendency to form fusion intermediates and render an uncleaved prefusion-optimized trimer. To this end, we tested two gp140 trimers based on HR1 redesign 1 and an NFL-like linker ($2 \times G_4S$)[30] (Fig. 5a). These two constructs, termed cleavage site long (CSL) and CSL-SOS, were transiently expressed in HEK293F cells followed by GNL purification and SEC on a Superdex 200 16/600 column.

Both CSL trimers showed similar expression levels and SEC profiles to those of CSF trimers (Figs 5b and 4b). For CSL, although no extra bands were definitively identified on the BN gel, the trimer bands appeared to be more diffuse than those observed for CSL-SOS (Fig. 5c). To further characterize their structures, we obtained negative-stain EM reconstructions for the unliganded CSL and CSL-SOS trimers at $\sim 17$ and 20 Å resolutions, respectively (Fig. 5d and Supplementary Fig. 10). The CSL trimer showed a somewhat different morphology than WT SOSIP, HR1 redesign 1, prefusion CSF and CSF-SOS trimers: the density of CSL trimer appeared to be narrower at the top of the trimer apex with additional densities pointing outwards and a wider bottom around gp41$_{ECTO}$. The overall shape of the CSL-SOS trimer was consistent with that of the CSF-SOS trimer. The $\sim 20$-Å reconstructions of PGV04-bound CSL and CSL-SOS trimers resembled that of the SOSIP trimer, indicative of stabilization on bNAb binding (Supplementary Fig. 4e). Taken together, our results suggest that a long cleavage site linker can reduce the formation of fusion intermediates.

We performed antigenic profiling for the CSL and CSL-SOS trimers using the same panel of bNAbs and non-NAbs as for the two HR1 redesigns (Fig. 6 and Supplementary Fig. 11). For apex-directed bNAbs PGDM1400, PG16 and PGDM145, the two CSL trimers showed similar binding kinetics and affinities to those of HR1 redesign 1. For CD4bs-directed bNAb VRC01, NAb b12, and glycan-reactive bNAbs (PGT121, PGT128, PGT135 and 2G12), CSL and CSL-SOS showed nearly identical binding profiles to those of HR1 redesign 1 and the SOSIP trimer. As expected, the most visible difference was found for bNAbs PGT151 and 35O22. For PGT151, which binds an epitope consisting of one gp120 and two adjacent gp41s in a trimer[18,19], the cleaved HR1-redesigned trimers and SOSIP trimer showed flat dissociation curves (Fig. 3c). However, the CSL trimers showed faster off-rates similar to those observed for the CSF trimers, indicating a consistent effect caused by cleavage site linkers (Fig. 6a). By contrast, the off-rates appeared to be cleavage-independent for 35O22, which targets gp120 and gp41 in a single gp140 protomer[21]. Binding to F240 and 7B2 revealed less accessible, non-neutralizing epitopes on gp41 for CSL-SOS but not for CSL, consistent with observations for the two CSF trimers (Fig. 6b).

In summary, an NFL-like long linker between gp120 and gp41 used in combination with an optimal HR1 redesign yielded an uncleaved gp140 that retained the most desirable traits of a prefusion trimer. Our comparative analysis of linker length also revealed complex consequences of cleavage site modification. Therefore, changing the linker length at the cleavage site must be carefully evaluated in each case of trimer immunogen design.

**A generic HR1 linker to stabilize diverse Env trimers**. Although cleavage site linkers might cause complications, HR1 redesign appeared to have an overall positive effect on trimer structure and antigenicity. In light of this finding, we revisited the strategy for HR1 redesign by examining the utility of a simple GS linker (Fig. 7a). If such a 'generic' HR1 linker (termed HR1-G) is proven successful, it will not only confirm the role of this HR1 region in Env metastability but also enable further development of stable trimers for diverse HIV-1 strains. We first tested the generic HR1 linker in the context of clade-A BG505, clade-B JRFL and clade-C DU172.17, with their WT SOSIP trimers included for comparison. For the three strains studied, the GS linker showed consistent improvement on trimer yield and purity (Fig. 7b). The most substantial improvement was observed for a clade-C strain: 46% increase of trimer peak relative to SOSIP with the aggregate and dimer/monomer peaks reduced by 34% and 37%,

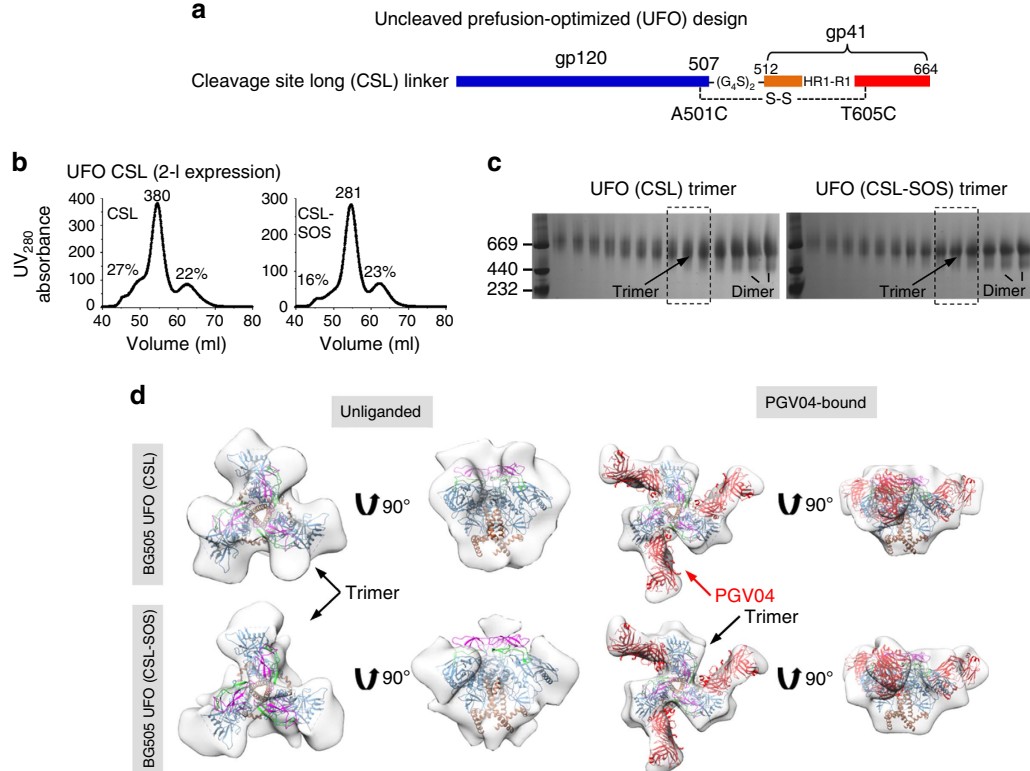

**Figure 5 | Design and characterization of HR1-redesigned Env trimers with long cleavage site linkers.** (**a**) Schematic representation of CSL linkers. The resulting trimers are termed uncleaved prefusion-optimized (UFO) trimers. (**b**) SEC profiles of CSL and CSL-SOS trimers from a Superdex 200 16/600 column. The UV value of the trimer peak and the percentages for UV values of aggregate peak (at 50 ml) and dimer/monomer peak (at 62.5 ml) relative to the trimer peak (at 55 ml) are labelled for comparison. (**c**) BN-PAGE of 293F-expressed, GNL-purified CSL and CSL-SOS trimers following SEC separation on a Superdex 200 16/600 column. The fractions used for EM analysis and antigenic profiling are circled by black dotted lines with expected positions for trimer and dimer bands labelled on the gel. (**d**) 3D reconstructions of CSL and CSL-SOS trimers in unliganded and PGV04-bound forms derived from negative-stain EM. The trimer densities are shown in grey transparent surface with the high-resolution cryo-EM structure of the SOSIP trimer (PDB 3J5M, gp120 in blue with V1V2 in magenta, V3 in green, gp41 in brown and PGV04 Fab in red). Both top and side views are shown after fitting the previously published EM model (PDB ID: 3J5M) into the density.

respectively. For this clade-C strain, two top-ranking HR1 redesigns from the ensemble-based protein design further increased the trimer peak by ∼50% with identical SEC profiles to HR1-G. For the two B′/C recombinant strains, CH115.12 and CN54, similar SEC profiles with a less pronounced improvement in trimer purity were observed for the HR1-redesigned and CSF-SOS trimers, suggesting additional factors contributing to the metastability of B′/C recombinant Envs (Fig. 7c). Overall, the results indicate that the generic HR1 linker offers a general framework for stabilization of Env while further improvement of trimer properties can be achieved by computational design in a strain-specific manner (Fig. 7d).

## Discussion

An effective HIV-1 vaccine will likely involve immunogens that mimic the native, prefusion trimer[47]. However, development of such trimer immunogens has been hampered by the labile nature of the Env spike and, until recently, the lack of structural details. This obstacle was spectacularly overcome by the BG505 SOSIP.664 gp140 trimer[11], which facilitated high-resolution analyses of Env structure[14–16,35] and provided a rational basis for vaccine design. Recent advances include expansion of the SOSIP design to other HIV-1 strains[23,24], incorporation of new stabilizing mutations[26] and removal of furin dependency by using cleavage site linkers[27,29,30]. However, a premium is placed on trimer purification to remove unwanted Env forms and

misfolded trimers. Trimer purification is currently achieved by complex methods such as bNAb affinity column, negative selection and multi-cycle SEC, which can be adapted for industrial-scale production but will likely require special considerations. It is plausible that trimer impurity and production inefficiency are linked to the fundamental causes of Env metastability that have not been completely resolved by previous trimer designs.

In this study, we investigated the underlying causes of HIV-1 Env metastability with a rational design strategy. We first directed our attention to an HR1 region that undergoes drastic conformational change during viral fusion with host cells. This region (residues 547–569) has not been explicitly explored, with the exception of the I559P mutation in the SOSIP design[10] that was originally speculated to destabilize the postfusion state[10] and validated by recent structural studies[35,36]. We shortened and computationally optimized this region in an attempt to stabilize the prefusion trimer conformation. Significantly, almost all HR1 redesigns showed substantial improvement in trimer yield and purity, suggesting that this disordered HR1 region is a major site of metastability and may indeed be a source of folding inefficiency and sample impurity in the production of soluble Env trimers. Structural analysis and antigenic profiling together established HR1 redesign 1 as a primary candidate to investigate the effect of cleavage site modification. While linker length has been extensively examined for NFL and sc-gp140 trimers[29,30], its effect on trimer metastability has not been addressed. The major effort

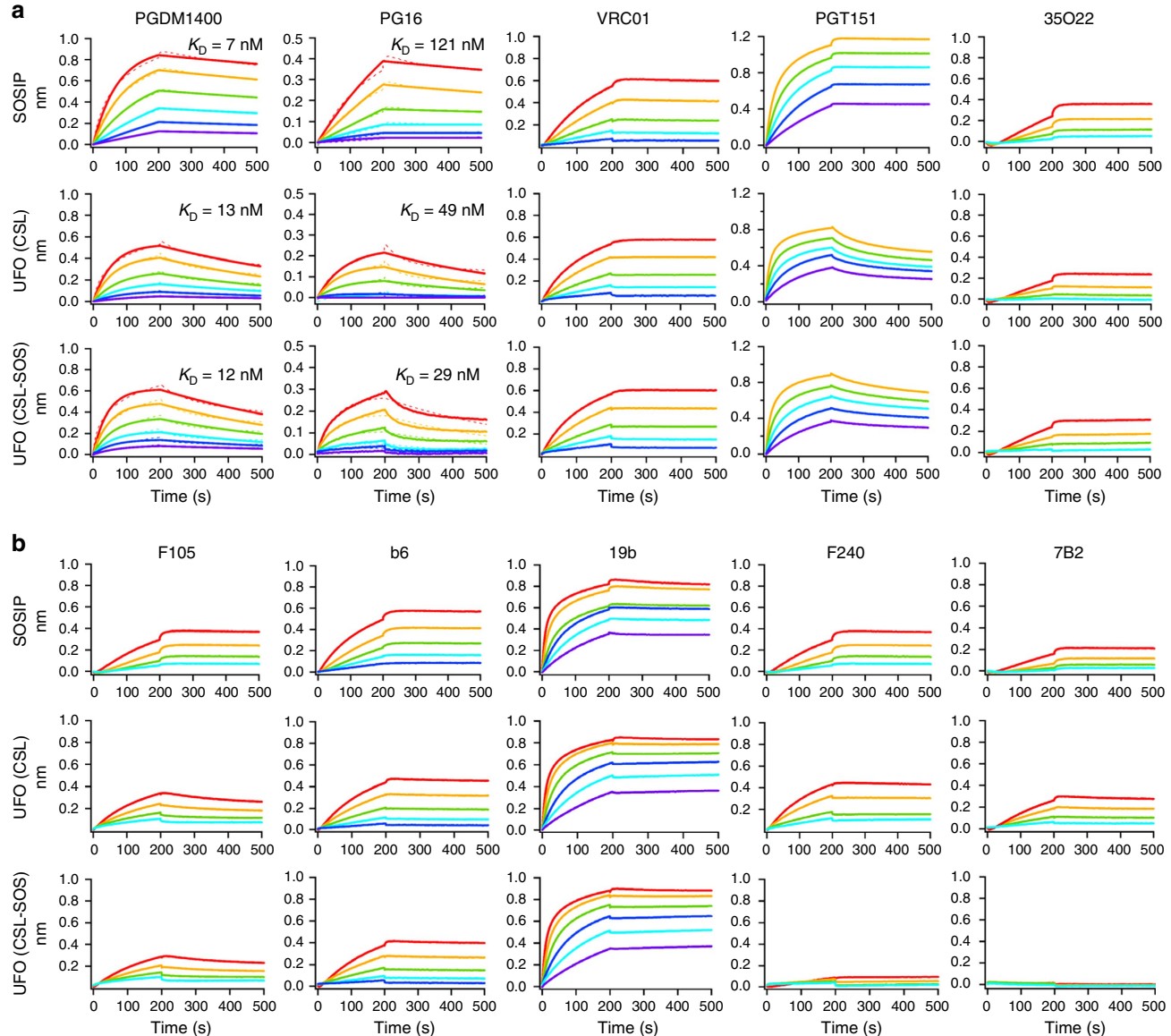

**Figure 6 | Antigenic profiles of UFO trimers.** Antibody-binding kinetics were measured for CSL and CSL-SOS trimers using a panel of representative (**a**) bNAbs and (**b**) non-NAbs, with additional antibody binding profiles shown in Supplementary Fig. 11. Sensorgrams were obtained from an Octet RED96 using a trimer titration series of six concentrations (200–12.5 nM by twofold dilution). $K_D$ values calculated from 1:1 global fitting are labelled for V1V2 apex-directed bNAbs (PGDM1400 and PG16) in **a**. UFO, uncleaved prefusion-optimized.

in those studies had been to remove unwanted Env species resulting from cleavage site modification rather than to determine their cause. We examined various cleavage site linkers in the context of an HR1 redesign. In the uncleaved gp140 form, the tendency for Env misfolding appeared to be inversely correlated with linker length, as revealed by the analysis of CST, CSF and CSL trimers. By collapsing the trimer ensemble into two predominant states, the CST and CSF designs captured possible fusion intermediates that are most likely inherent to uncleaved gp140 trimers but were previously inaccessible. Future high-resolution studies of such intermediate states could greatly enhance our understanding of the HIV-1 fusion mechanism.

The results from a generic HR1 linker have important implications. The high trimer yield and purity obtained from this simple GS linker and further improvement by computational design suggested a general strategy for identifying vaccine-suitable strains and designing stable gp140 trimers based

on these strains. Finally, given the presence of an HR1-like region in most enveloped viruses using the type 1 fusion mechanism[48–50], the generic HR1 redesign is potentially a universal strategy to stabilize soluble envelope trimers for vaccine development.

## Methods

**Ensemble-based _de novo_ protein design.** We developed an ensemble-based _de novo_ protein design method (Fig. 1c). Given the Env trimer structure (PDB ID: 4TVP) and a specified loop length, a three-step design process was undertaken: (1) an ensemble of backbone conformations (1,000) is generated to connect the two anchor residues using a torsion-space loop sampling algorithm[51]; (2) for each loop backbone, a starting sequence is selected from a pool of 50 random sequences based on the RAPDF potential[52] and subjected to 500 steps of Monte Carlo simulated annealing with the temperature linearly decreasing from 300 to 10 K; (3) the lowest-energy sequence for each backbone is recorded and all Monte Carlo simulated annealing-derived designs are ranked based on energy at the completion of the process. The top 20 designs are then manually inspected to facilitate selection of 5 candidates for experimental validation. The computer code for ensemble-based protein design can be obtained from J.Z. upon request.

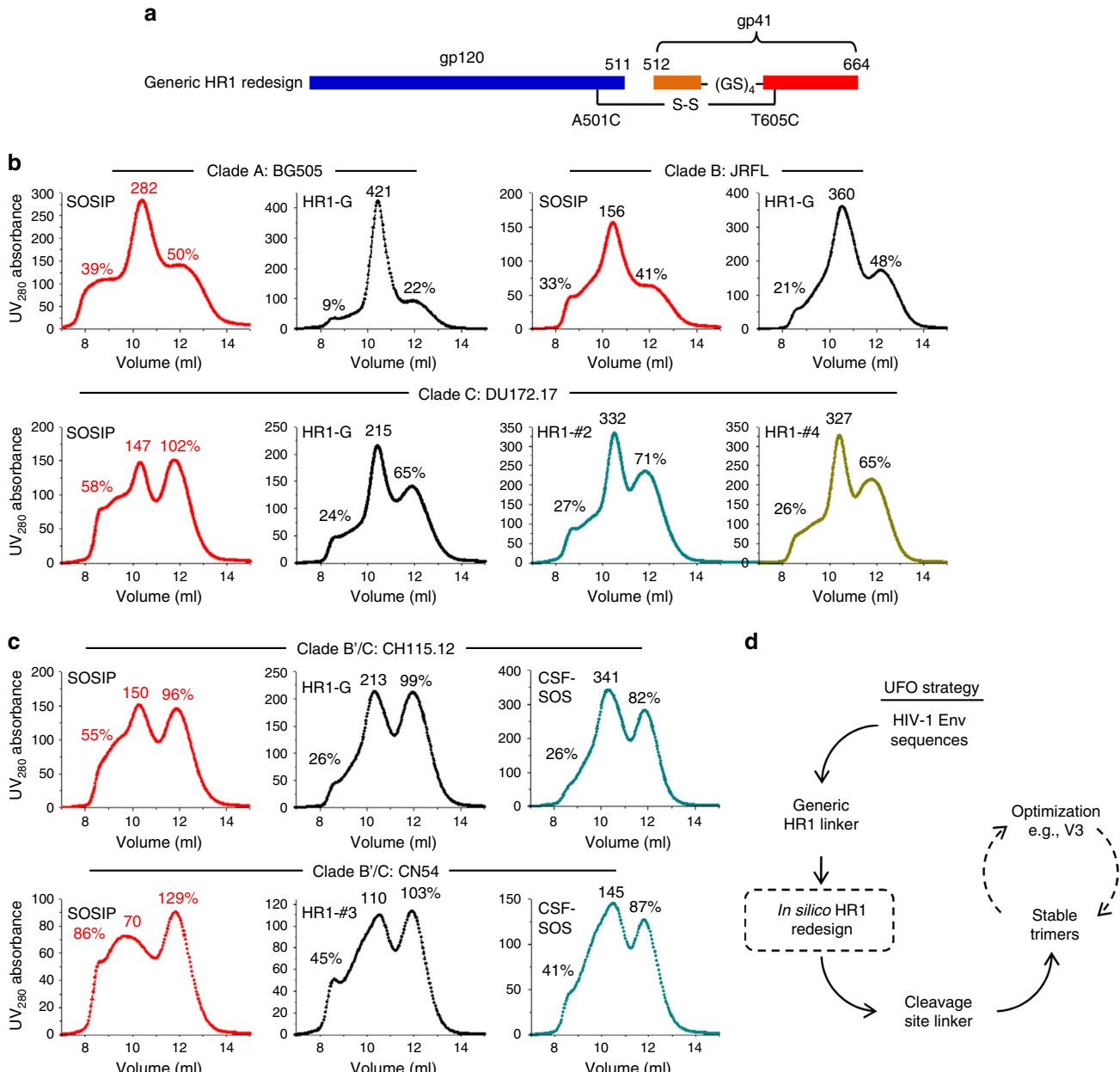

**Figure 7 | Design and validation of a generic HR1 linker to stabilize diverse Env trimers.** (**a**) Schematic representation of a generic HR1 linker (HR1-G). (**b**) SEC profiles of WT SOSIP.664 and HR1-G trimers for clade-A BG505 (top left), clade-B JRFL (top right) and clade-C DU172.17 (bottom left) from a Superdex 200 10/300 column. For the clade-C strain, two top-ranking HR1 redesigns (#2 and #4, bottom right) from ensemble-based *de novo* protein design are also included for comparison. (**c**) SEC profiles of WT SOSIP.664, HR1-redesigned and CSF-SOS trimers for B′/C recombinant CH115.12 (top) and CN54 (bottom) from a Superdex 200 10/300 column. For CN54, a top-ranking HR1 redesign (#3) from ensemble-based *de novo* protein design is used as the basis for CSF-SOS. The UV value of the trimer peak and the percentages of UV values for aggregate peak (at 9 ml) and dimer/monomer peak (at 12 ml) relative to the trimer peak (at 10.5 ml) are labelled for comparison. (**d**) Schematic flow of the uncleaved prefusion-optimized (UFO) strategy to facilitate the selection of stable Env trimers for vaccine design (optional steps are indicated with dashed lines).

**Antibodies.** We utilized a panel of bNAbs and non-NAbs to characterize the antigenicity of redesigned trimers. 2G12 and b12, as well as non-NAbs F240, 7B2, 17b and A32, were requested from the NIH AIDS Reagent Program (https://www.aids-reagent.org/). Other bNAbs and non-NAbs were provided by D.S. and D.R.B.

**Expression and purification of HIV-1 Env trimers.** Env trimers were transiently expressed in HEK293F cells (Life Technologies, CA) except for crystallographic analysis. Briefly, 293F cells were thawed and incubated with FreeStyle 293 Expression Medium (Life Technologies, CA) in a shaker incubator at 37 °C, with 135 r.p.m. and 8% $CO_2$. When the cells reached a density of $2.0 \times 10^6 \, ml^{-1}$, expression medium was added to reduce cell density to $1.0 \times 10^6 \, ml^{-1}$ for transfection with polyethyleneimine (PEI) (Polysciences, Inc). For SOSIP and HR1-redesigned trimers,

800 μg of Env plasmid and 300 μg of furin plasmid in 25 ml of Opti-MEM transfection medium (Life Technologies, CA) was mixed with 5 ml of PEI-MAX ($1.0 \, mg \, ml^{-1}$) in 25 ml of Opti-MEM, whereas for uncleaved trimers, 900 μg of Env plasmid was used without furin. After incubation for 30 min, the DNA-PEI-MAX complex was added to 1-l 293F cells. Culture supernatants were collected 5 days after transfection, clarified by centrifugation at 1,800 r.p.m. for 20 min, and filtered using 0.45-μm filters (Thermo Scientific). The Env proteins were extracted from the supernatants using a GNL column (Vector Labs). The bound proteins were eluted with PBS containing 500 mM NaCl and 1 M methyl-α-D-mannopyranoside and then purified by SEC on a Superdex 200 Increase 10/300 GL column for initial assessment, and a HiLoad 16/600 Superdex 200 PG column (GE Healthcare) for EM analysis and antigenic profiling. Protein concentrations were determined using $UV_{280}$ absorbance with theoretical extinction coefficients.

**BN-PAGE.** Env proteins were analysed by BN-PAGE and stained with Coomassie blue. The protein samples were mixed with G250 loading dye and added to a 4–12% Bis-Tris NuPAGE gel (Life Technologies). BN-PAGE gels were run for 2 h at 150 V using the NativePAGE running buffer (Life Technologies) according to the manufacturer's instructions.

**Differential scanning calorimetry.** Thermal melting curves of SOSIP and HR1-redesigned trimers were obtained with a MicroCal VP-Capillary calorimeter (Malvern). The SEC-purified glycoproteins were buffer exchanged into $1 \times$ PBS and concentrated to 0.5–1 µM before analysis by the instrument. Melting was probed at a scan rate of $90\,^{\circ}\text{C}\,\text{h}^{-1}$. Data processing including buffer correction, normalization and baseline subtraction was conducted using the standardized protocol from the Origin 7.0 software.

**Protein production and purification for crystallization.** The two HR1-redesigned trimers, as well as the Fabs 8ANC195 and PGT128, were produced and purified as previously described[38]. Briefly, all constructs were cloned into the expression vector phCMV3. Fabs were transiently transfected into mammalian FreeStyle 293F cells, and trimers were transiently transfected in GnTI$^{-/-}$ 293S cells. One week after transfection, the supernatants of the antibody-transfected cells were collected and purified using a LC-λ and a LC-κ Capture Select column (BAC BV) for PGT128 and 8ANC195, respectively, before further purification by ion exchange chromatography and SEC. The supernatants of the trimer-transfected cells were purified using a 2G12 affinity column followed by SEC. The trimer complexes used for crystallization trials were prepared by mixing the trimer proteins with Fabs PGT128 and 8ANC195 at a molar ratio of 1.0:1.2:1.2 at room temperature for 20 min. This mixture was then partially deglycosylated using endoglycosidase H (EndoH) in 200 mM NaCl, 50 mM sodium citrate, pH 5.5, 37 °C for 1 h following the manufacturer's protocol (New England Biolabs) before final purification by SEC.

**Protein crystallization and data collection.** Two purified protein complexes containing Fabs PGT128 and 8ANC195 bound to HR1-redesigned trimers were prepared for crystallization by buffer exchange into 50 mM NaCl, 20 mM Tris-HCl, pH 7.2. The complexes were then concentrated to 5 mg ml$^{-1}$ and passed through a 0.22-µm filter before crystal screening using the IAVI/JCSG/TSRI CrystalMation robot (Rigaku) at the JCSG[53]. Similar to a previously described complex of Fabs PGT128 and 8ANC195 with a BG505 SOSIP gp140 trimer[38], the HR-1 redesigned trimer/Fab complexes were crystallized at 25 °C in 0.05 M lithium sulfate, 0.05 M sodium sulfate, 20% (w/v) PEG 400 and 0.05 M Tris-HCl, pH 8.7 (JCSG Core Suite 1 condition: A03, Qiagen). All crystals for X-ray data collection were cryoprotected by brief immersion in mother liquor supplemented with 40% PEG 400 before flash-cooling in liquid nitrogen. For the HR1 redesign 1 complex, diffraction data to 6.3 Å resolution were collected at beamline 23ID-B at the Advanced Photon Source, processed with HKL-2000 (ref. 54), and indexed in space group I23 with 100% completeness and an $\langle I \rangle / \langle \sigma_I \rangle$ of 2.3 in the highest-resolution shell. For the HR1 redesign 9 complex, diffraction data to 6.9 Å resolution were collected at beamline 12-2 at the Stanford Synchrotron Radiation Lightsource, processed with HKL-2000 (ref. 54), and indexed in space group I23 with 100% completeness and an $\langle I \rangle / \langle \sigma_I \rangle$ of 1.3 in the highest resolution shell. Data collection and processing statistics are summarized in Table 1.

**Structure determination and refinement.** The HR1-redesigned gp140 trimer structures bound to PGT128 and 8ANC195 were solved by the molecular replacement method using Phaser[55] and a search model consisting of a BG505 SOSIP.664 Env trimer bound to Fabs PGT128 and 8ANC195 (PDBID: 5C7K). As described previously[38], refinement consisted of alternating rounds of manual model building using Coot-0.7 (ref. 56) and automated refinement as implemented by the Phenix programme[57]. Given the limited resolution of the data sets, grouped B-factor refinement for each residue was used. Furthermore, positional coordinate refinement was enforced using a reference model set of restraints. The starting model for each automated refinement session in Phenix was defined as the reference model for that session. Finally, the model was minimally modified except at the HR1 site of redesign. The final $R_{\text{cryst}}$ and $R_{\text{free}}$ values converged at 28.1% and 32.2%, and 28.4% and 32.2% for the complex structures of HR1 redesigns 1 and 9, respectively. Coordinate error was calculated by the Phenix refinement package using a maximum likelihood (ML)-based method[57,58] (see Table 1 for final refinement statistics). Fab residues were numbered according to Kabat nomenclature[59] and gp140 was numbered using the standard HXBc2 convention. The redesigned regions in HR1 redesigns 1 and 9 were numbered 547A-H and 547A-J, respectively.

**EM sample preparation.** The Env gp140 trimers alone, and in complex with Fab PGV04, were analysed by negative-stain EM. A 3-µl aliquot containing $\sim 0.02$ mg ml$^{-1}$ of the trimers was applied for 15 s onto a carbon-coated 400 Cu mesh grid that had been glow discharged at 20 mA for 30 s, then negatively stained with 2% uranyl formate for 30 s. Data were collected using a FEI Tecnai Spirit electron microscope operating at 120 kV, with an electron dose of $\sim 30\,\text{e}^{-}\,\text{Å}^{-2}$

and a magnification of $\times 52,000$ that resulted in a pixel size of 2.05 Å at the specimen plane. Images were acquired with a Tietz $4k \times 4k$ TemCam-F416 CMOS camera using a nominal defocus of 1,000 nm and the Leginon package[60].

**Image processing and 3D reconstruction.** Particles were selected automatically from the raw micrographs (Supplementary Fig. 12) using DoG Picker and put into a particle stack using the Appion software package[61]. Initial, reference-free, two-dimensional (2D) class averages were calculated using particles binned by two via iterative multivariate statistical analysis/multireference alignment and sorted into classes[62]. Particles corresponding to trimers or to trimers bound to PGV04 were selected into a substack and binned by two before another round of reference-free alignment was carried out using the iterative multivariate statistical analysis/multireference alignment and Xmipp Clustering and 2D alignment programmes[63]. To analyse the quality of the trimers (closed native-like, partially open native-like and non-native), the reference-free 2D class averages were examined by eye using the metrics previously described[23]. An ab initio common lines model (Supplementary Fig. 13) was calculated from reference-free 2D class averages in EMAN2 (ref. 64) imposing symmetry C3. This model was then refined against raw particles for an additional 25 cycles using EMAN[65]. The resolutions of the final models were determined using a Fourier Shell Correlation cutoff of 0.5. The number of particles collected and used for the final 3D reconstructions is summarized in Supplementary Table 1. For the CSF reconstructions (states 1 and 2), after 2D classification, unbinned particles were refined against a density map of the unliganded trimer (Supplementary Fig. 13) filtered to a 60-Å resolution. 3D classification was then performed in RELION version 1.4b resulting in six classes within each data set[66,67]. Stable classes were selected for further processing followed by various steps of additional 3D classification. Stoichiometric classes were selected for further refinement to generate the final maps for states 1 and state 2, respectively. The EM maps were contoured at different threshold values to facilitate a thorough comparison of the obtained 3D reconstructions (Supplementary Fig. 14).

**Bio-layer interferometry.** The kinetics of trimer binding to bNAbs and non-NAbs was measured using an Octet Red96 instrument (fortéBio, Pall Life Sciences). All assays were performed with agitation set to 1,000 r.p.m. in fortéBIO $1 \times$ kinetic buffer. The final volume for all the solutions was 200 µl per well. Assays were performed at 30 °C in solid black 96-well plates (Geiger Bio-One). A unit of 5 µg ml$^{-1}$ of antibody in $1 \times$ kinetic buffer was loaded onto the surface of anti-human Fc Capture Biosensors (AHC) for 300 s. A 60-s biosensor baseline step was applied before the analysis of the association of the antibody on the biosensor to the Env trimer in solution for 200 s. A twofold concentration gradient of trimer starting at 200 nM was used in a titration series of six. The dissociation of the interaction was followed for 300 s. Correction of baseline drift was performed by subtracting the averaged shift recorded for a sensor loaded with antibody but not incubated with trimer, or a sensor without antibody but incubated with trimer. Octet data were processed by fortéBio's data acquisition software v.8.1. Experimental data were fitted for V1V2 apex-directed bNAbs using a global fit 1:1 model to determine the $K_D$ values and other kinetic parameters.

**Data availability.** The authors declare that all data supporting the findings of this study are available within the article and its Supplementary Information files. Coordinates and structure factors for HR1 redesigns 1 and 9 are deposited with the RCSB protein data bank (PDB) under accession codes 5JS9 and 5JSA, respectively. The negative-stain EM reconstructions are deposited in the electron microscopy data bank (EMDB) under accession codes EMD-6590 (CSF state 1), EMD-6591 (CSF state 2), EMD-6592 (CSF state 1 bound to PGV04), EMD-6593 (CSF state 2 bound to PGV04), EMD-6621 (CSF-SOS), EMD-6622 (CSF-SOS bound to PGV04), EMD-6623 (CSL), EMD-6634 (CSL bound to PGV04), EMD-6625 (CSL-SOS), EMD-6626 (CSL-SOS bound to PGV04), EMD-6627 (HR1 redesign 1) and EMD-6628 (HR1 redesign bound to PGV04).

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

## Acknowledgements

We are very grateful to R. Stanfield for helpful discussions, M. Elsliger for computer support and H. Tien for crystallization screening; X-ray data sets were collected at the Advanced Photon Source (APS) beamline 23ID-B and Stanford Synchrotron Radiation Lightsource (SSRL) BL12-2. Use of the APS was supported by the US Department of Energy, Basic Energy Sciences, Office of Science, under contract no. DE-AC02-06CH11357. SSRL is a Directorate of SLAC National Accelerator Laboratory

and an Office of Science User Facility operated for the U.S. Department of Energy Office of Science by Stanford University. The SSRL Structural Molecular Biology Program is supported by the DOE Office of Biological and Environmental Research, and by the NIH, National Institute of General Medical Sciences (including P41GM103393) and the National Center for Research Resources (P41RR001209). Electron microscopy data were collected at the Scripps Research Institute EM Facility. This work was supported by the International AIDS Vaccine Initiative Neutralizing Antibody Center and CAVD (OPP1084519 and OPP1115782), by the Center for HIV/AIDS Vaccine Immunology and Immunogen Discovery (CHAVI-ID UM1 AI100663) (A.B.W., I.A.W., J.Z. and D.R.B.), by the HIV Vaccine Research and Design (HIVRAD) programme (P01 AI110657) (A.B.W. and I.A.W.), by the Joint Center of Structural Genomics (JCSG) funded by the NIH NIGMS, Protein Structure Initiative (U54 GM094586) (I.A.W.), and AI084817 (I.A.W., A.B.W.). A portion of this work was supported by an American Foundation for AIDS Research Mathilde Krim Fellowship in Basic Biomedical Research (L.K.).

## Author contributions

L.K., L.H., N.d.V., A.B.W., I.A.W. and J.Z. designed research; L.K., L.H., N.d.V., N.V., C.D.M., P.A., B.Z. and J.Z. performed experiments; D.S. and D.R.B. provided bNAbs PG16, PGT145, PGDM1400, PGT121, PGT128, PGT135 and PGT151, as well as non-NAbs b6, F105 and 447-52D; L.K., L.H., N.d.V., N.V., C.D.M., P.A., B.Z., A.B.W., I.A.W. and J.Z. analysed the data; L.K., L.H., N.d.V., N.V., C.D.M., A.B.W., I.A.W. and J.Z. wrote the paper. All authors were asked to comment on the manuscript. This is TSRI manuscript number 29247.

## Additional information

**Competing financial interests:** The authors declare no competing financial interests.

