## [Peer review file · Nature Communications]

REVIEWERS' COMMENTS:

Reviewer #3 (Remarks to the Author):

Jiang Zhu and colleagues have adequately addressed the comments/concerns raised by this reviewer and others and have improved the significance and clarity of their manuscript. My view is that this work is now appropriate for publication in Nature Communications.

Point-by-Point Response to Review Comments

Reviewer 1:

(A) Summary of the key results: Kong et al. provides a highly-accessible, "cookbook" format roadmap for generalized protocols for producing soluble HIV Env proteins in forms that largely recapitulate the structural and antigenic properties of native Env proteins - as well as a detailed description of the process, highlighting pitfalls and workarounds. Consideration was duly given to the limitations of general laboratory resources, with an end result of a highly transportable protocol. Discussion of the meaning of the biochemistry and biophysical results in context of the larger goals of the research was exemplary, and greatly appreciated. Therefore, the manuscript as written delivers tremendous value to the community.

Response:

We appreciate the reviewer's positive comments. Although our primary goal in this study was to identify the underlying causes of HIV Env metastability by testing various gp140 designs, it was our hope that through this process we could establish a general protocol for producing soluble HIV Env trimers using materials accessible to most laboratories. We are very grateful to the reviewer for recognizing our effort and the potential value of such a protocol to the community. We expect that the UFO trimer, with improved yield and purity, will lead to further opportunities for HIV vaccine development, in particular, the use of nucleic acid-based delivery platforms, for which Env proteins including the trimer would be produced by host cells without the need for cleavage by furin. Based on the reviewer's comments, we have modified the X-ray structure table (Table S1) and SPR figure (Fig. S6). To address other reviewers' comments, we have included new data for another B'/C recombinant strain (CN54, Fig. 7c), a schematic flow of the UFO design strategy (Fig. 7d), and more technical details of the EM analysis in the Methods and supplemental figures.

(B) Originality and interest: if not novel, please give references: While building on a wide foundation of previous results, the synthesis of the results presented in this manuscript is quite novel and of direct interest to the community. The extensive presentation and discussion of the results throughout the process also becomes a useful guide for applying rigorous biochemistry in these contexts.

(C) Data & methodology: validity of approach, quality of data, quality of presentation: Methods and results are clearly presented and thoroughly described. Data validation and quality are completely appropriate and fully support the conclusions as formulated.

(D) Appropriate use of statistics and treatment of uncertainties: While most of the results are qualitative or semi-quantitative in nature, error estimates are spotty: for example, no estimate of overall coordinate error is provided in Table S1 (very useful at these resolutions), and errors are unevenly reported in Figure S6. Also, "29.3 (plus minus) 0.05 RU" means that the value was truncated one significant digit too early.

Response:

We have now reported the overall coordinate errors of both crystal structures in Table S1, and corrected the K_D value in Fig. S6. Of note, Scrubber software does not provide standard deviations. In this study, the Scrubber output was included to confirm the results obtained from BIA evaluation software.

(E) Conclusions: robustness, validity, reliability: The overall impression generated is that the data quite exhaustively support the conclusions and provide considerable internal validation - likely highly reliable.

(F) Suggested improvements: experiments, data for possible revision: It's not necessary to quote lattice angles for cubic space groups in Table S1.

Response:

Based on the reviewer's suggestion, we have now removed the lattice angles in Table S1.

(G) References: appropriate credit to previous work? Literature citations are very complete and thorough.

(H) Clarity and context: lucidity of abstract/summary, appropriateness of abstract, introduction and conclusions: Extremely well-written; my recommendation is to publish as is.

Reviewer 2:

General comments: The manuscript by Kong et al discusses a comprehensive, multipronged approach to stabilize soluble trimeric forms of the HIV envelope glycoprotein. It follows reports on the design of stabilized forms of this viral protein by several groups (SOSIP or NFL). The overall goals of the present study is to identify additional modifications that would improve the stability of soluble trimers in their pre-fusion conformation, they will improve the relative expression of trimers over aggregates and other envelope species (monomers/dimers) during mammalian expression and will improve the antigenic profile of these soluble proteins to better mimic the configuration of the function envelope trimer on the surface of virions (occlusion of non-neutralizing epitope, but exposure of neutralization epitopes). Three major approaches were evaluated: (i) replacing the HR1N domain of gp41 (547-569) by 8 or 10 residues, (ii) replacing the furin cleavage site between gp120 and gp41 (either by a 8 residue linker or by SGS), (iii) replacing the furin cleavage site by longer linkers, and (iv) by a generic HR1 linker.

Each construct was evaluated by complementary methodologies: Expression and purification, thermostability, structural analysis (crystallography and/or EM) and antigenic profile (using broadly neutralizing and non-neutralizing antibodies). To my knowledge, this must be the most complete analysis of diverse soluble HIV envelopes. It is very impressive. Having said that, as it concerns the antigenic structure of these proteins, it does not appear that one redesign approach is clearly superior to the others. Also, regarding the HR1 generic redesign: why is this 'generic' since it benefits only a small fraction of backbones examined? It has no obvious benefit for example to the B/C clade envelope, or the clade B JRFL. Overall the study does not provide a universal way of stabilizing diverse trimers, something that would have been very exciting. Finally, the V3 loop remains well exposed on all these redesigns, as it is exposed on the original SOSIP/NFL designs. The implications of this exposure to the ability of stabilized trimers to elicit broadly neutralizing antibodies are well known to these groups. The above comments and concerns should be addressed by the reviewers.

Response:

We thank the reviewer for insightful comments and for acknowledging that our study presented the most complete analysis of diverse soluble HIV-1 envelopes. Indeed, based on the reviewer's comment, we have included another B'/C recombinant strain, CN54, in Fig. 7c (see below).

As demonstrated in previous studies, the BG505 SOSIP trimer is an excellent structural and antigenic mimic of the native-like, pre-fusion viral spike. It is perhaps not realistic to expect superior antigenicity from any new trimer platforms, as BG505 SOSIP was already excellent. However, we did observe reduced exposure of immunodominant epitopes within gp41 for the HR1-redesigned trimers and the UFO trimers, which was also noted by the reviewer. Our primary goal in this study was to probe the underlying causes of HIV Env metastability by testing various trimer designs. While the shortening of the HR1 N-terminus displayed overall positive impact on the trimer properties, the cleavage site linker resulted in more complications than previously appreciated. Although no single HR1 redesign 'dramatically' outperformed others, several variants did show notably high trimer yield and purity. The similar qualities of diverse HR1 redesigns (both computationally derived and generic) thus provided the most compelling

evidence for our hypothesis that HR1 N-terminus is the primary cause of metastability. Thus, once this bend at the top of the long HR1 helix was modified to eliminate its tendency to refold into a more extended helix, the trimer properties were consistently improved.

The 8-residue GS linker was considered 'generic' for several reasons. First, GS linkers are widely used by experimental biologists and thus provide a highly accessible platform to screen diverse HIV-1 envelopes; Second, this GS linker showed consistent improvement in trimer yield over the wild-type SOSIP design for all tested HIV-1 strains (clade-A BG505: 50% increase clade-B JRFL: 130% increase; clade-C DU172.17: 46% increase; and clade-B'/C: 42-57% increase), Third: this GS linker consistently reduced the aggregated fraction of the expressed Env protein for all tested HIV-1 strains. We agree with the reviewer that the GS linker did not dramatically alter the SEC profile for JRFL, for which SOSIP and NFL already yielded relatively high trimer purity. Of note, the GS linker did show the most visible improvement in trimer yield for JRFL, with a 130% increase compared to a ~50% increase for other strains. For CH115.12, we speculate that the less pronounced improvement may be specific for the B'/C recombinant strain. We examined this possibility by testing another B'/C recombinant strain, CN54, and observed similar SEC profiles. Taken together, although UFO presents a general platform for trimer stabilization, further modification may be required for certain specific strains. We have now included the SEC profiles for CN54 (Fig. 7c) and added a schematic figure (Fig. 7d) to illustrate the UFO strategy.

As stated previously, this study was focused on the primary causes of HIV-1 Env metastability and was certainly not intended to solve the inherent vaccine problems all at once. But we thank the reviewer for commentary on the V3 exposure and its implications for the ability of stabilized trimers to elicit bNAbs in immunization. We cautiously expect that the improved trimer properties brought about by the UFO design will allow other modifications (e.g. V3 stabilization) to be incorporated into the trimer construct to achieve desirable *in vivo* outcomes, as illustrated now for other redesigned Env trimers. This possibility will be explored in our follow-up studies. We have now included the V3 optimization in the schematic flow of the UFO strategy (Fig. 7d).

(1) Minor comment 1: On the backbone of SOSIP BG505, they replaced the HR1N domain of gp41 (547-569) by of 8 or 10 residues, in an effort to stabilize the pre-fusion conformation of the trimeric envelope spike. The two redesigns, appear to reduce exposure of some gp41 non-neutralization epitopes. This is interesting. Pg 12, first paragraph: 'while all three trimers (SOSIP, redesign 1 and redesign 9) showed some V 3 exposure.' I would say that based on the nm shifts shown in Fig 3c/d, the V3 is pretty well exposed on these proteins. Even better than the epitopes of VRC01, for example. Therefore, the text should be modified.

Response:

The reduced exposure of gp41 non-neutralizing epitopes is consistent with the fact that the HR1 bend is located in gp41 and will undergo dramatic and irreversible conformational change to transform gp41 into an extended helix during receptor and coreceptor binding and subsequent fusion. Therefore, it is not so surprising that, although the HR1 redesigns improve trimer yield, purity, and gp41 stability, it had little effect on the exposure of V3 loop, which is located at the apex of the trimer quite far from the HR1 modifications.

We agree with the reviewer that the V3 loop appeared to be well exposed from the BLI measurements (Fig. 3c/d), which is why we validated the BLI results by surface plasmon resonance (SPR) using the 2G12-purified SOSIP trimer. Indeed, our SPR experiment showed binding profiles and RU values (~30) similar to those reported for the BG505.664 SOSIP trimer in the Yasmeen et al. paper (Retrovirology 11:41, 2014), confirming the quality of the trimers tested. We also speculate that differences in the methods (BLI vs. SPR) may have contributed to the more visible V3 exposure by BLI, but not by SPR. In

brief, a small portion of native-like trimers may expose the V3 loop due to inherent flexibility and give rise to the observed 19b binding.

Because the CD4bs is recessed between two gp140 protomers, it is perhaps not surprising that the VRC01 binding to native-like SOSIP and UFO trimers was not as pronounced as the 19b binding to V3. As stated in our response to the reviewer's general comment, we expect that further modification in and around the V3 loop in an HR1-redesigned trimer or a UFO trimer will reduce V3 exposure and lead to more desirable antigenicity/immunogenicity and perhaps vaccine outcomes.

Based on the reviewer's suggestion, we have modified the text, on page 11, and included an optimization step (to reduce V3 exposure) in the schematic flow of the UFO strategy (Fig. 7d).

(2) Minor comment 2: CST or CSF redesigns: the label on the y axis of Sup Fig 9 need to change to correspond to CST and CSF (rather than CSF and CSF-SOSIP):

Response:

There might be a slight misunderstanding. For the uncleaved trimers, we tested three different designs of cleavage site linker, CST (T: truncated, without the disulfide bond SOS), CSF and CSF-SOS (F: near full-length, without and with the disulfide bond SOS), and CSL and CSL-SOS (L: long, without and with the disulfide bond SOS). We did not test antibody binding for the CST designs because they all showed an extra band on the BN gel, corresponding to a potential fusion intermediate that was also observed for the CSF trimer, for which we did indeed perform detailed structural and antigenic analyses.

We have now modified the Fig S9 legend to explain the difference between CSF and CSF-SOS.

Reviewer 3:

General comments: The HIV-1 envelope glycoprotein is a major target for immune recognition, however the development of effective vaccines requires stable immunogens that mimic the native pre-fusion trimer. This is challenging because of the instability of the trimer complex; thus a number of studies have focused on protein modifications to improve antigenic response. Kong et. al, have computationally re-designed the heptad region 1 of Env and replaced the gp120-gp41 cleavage site with different linkers with the goal of improving the yield of trimers that mimic the pre-fusion state. They use gel filtration chromatography, antigenic profiling and x-ray and negative-stain EM structural methods to compare the modified trimers and establish their approach for improving stability. Overall their computational design methods and altered cleavage site appear to enhance the stability of pre-fusion trimers and their structural biology approaches provide validation that the modified trimers are intact.

However, these findings overall do not seem significant enough to warrant publication in Nature Communications; a more methods-focused journal seems appropriate for this work. Furthermore, the significance and validation of the 'fusion intermediate' complex is somewhat unclear. I have only minor comments/questions regarding the EM characterization.

Response:

We appreciate the reviewer's positive comments on our computational design, improved trimer stability, and structural validation. Based on the reviewer's suggestions, we have revised the Methods, figure legends, and Supplemental Information to provide more technical details of the EM analysis.

We thank the reviewer for noting the 'fusion intermediate' state. It was a side effect of short cleavage-site linkers, but might have important implications. For example, a high-resolution structural analysis of this

construct will shed some light on the intermediate states as well as the transient epitopes displayed during the fusion process, which will inform future trimer design. Further, the impact of a ‘fusion intermediate’ on neutralizing B cell responses will also be of interest. One may theorize that such intermediates could potentially distract the immune system, thus dampening the bNAb responses. However, it is also possible that mixed pre-fusion and fusion-intermediate trimers may generate more diverse NAb clones with a broader coverage of epitopes presented on native virions. These hypotheses will be examined in our follow-up animal studies.

The goal of this study was to investigate the primary causes of HIV-1 Env metastability, which is at the core of HIV biology and vaccine design. We are confident that our findings are important and well in line with the scope of the journal that publishes research from all areas of the natural sciences.

(1) Why was cryo-EM not used? It seems for some of the trimer variants and PGV04-bound complexes this study would greatly benefit from higher resolution cryo-EM models. Given current technology and previous successes with the BG505 SOSIP complex this should be feasible with relatively small amounts of data. Negative-stain EM provides only low-resolution envelopes at ~20Å, which is minimal for the level of structural validation required for these studies.

Response:

While we agree with the reviewer that cryoEM would provide greater detail, the low resolution negative-stain EM data presented in the manuscript is sufficient to support the conclusions. We are indeed currently pursuing cryoEM studies of the CSF trimer in complex with PGV04 and expect to publish that data in a follow-up study.

(2) Please provide an example portion of a micrograph image and boxed-out single particles in order to assess the raw data.

Response:

Thanks- we have now added example micrographs with boxed-out particles in Fig. S12.

(3) Please provide information regarding the number of particles that went into each of the reconstructions - how many particles were collected and how many were used to generate the models.

Response:

We have now added a new table with this information (Table S2).

(4) How were the half-maps generated for determining the FSC? Was this using the currently accepted 'Gold Standard' method? Also, the resolution value by the FSC method is generally considered an estimate and not absolute. Please state that these are 'estimated' or 'indicated' resolutions.

Response:

We used the Gold standard method to determine the FSC at a value of 0.5 for the different maps. Based on the reviewer’s suggestion, we have also added estimates of the resolution in the main text and figure legends instead of an ‘absolute resolution’.

(5) Please indicate in the figure legends what the colors correspond to for the ribbon models that are docked into the EM maps, especially for PGV04. Indicating in the figures what the different lobes of density correspond to would also be helpful as well.

Response:

We have now indicated the colors that correspond to the ribbon models fitted into the EM densities in the figure legends. We have also added arrows in the figures to clarify the different components (Env trimer, Fab) in the EM densities.

(6) How were the two different EM maps for the BG505 CSF complex determined? 3D classification? How these two 'morphologies' were determined is not explained and leads to questions about whether the differences are real or an artifact.

Response:

We have now added further explanation in Methods and clarified that we used RELION 3D classification to derive the different reconstructions (pre-fusion and fusion-intermediate states).

(7) Please show the common lines initial model in the supplemental. Was one model used for all reconstructions? If different models were generated for each dataset then the differences in the final models could be due to model bias.

Response:

We have now added a new Fig. S13 with the common lines model used for all the refinements. We only used one common lines model to generate all the final maps, thus minimizing potential model biases.

(8) The reconstructions for the unliganded complexes in 4D and 5D seem poorly resolved with a lot of empty density that was unable to be docked with the structure. What was the rationale for the density threshold value that was used? A view of the maps at an increased threshold would be useful to see if the areas of higher density are centered with the model.

Response:

For Fab complexes, we typically choose contour levels that encompass the Fab. This typically works well for normalizing the variability in negative stain volume maps. For the unliganded complexes, it tends to be a bit more subjective. Thus we have now added a Fig. S14 with unliganded maps contoured at the same value as well as a higher threshold level.

Reviewers' comments:

Reviewer #1 (Remarks to the Author):

A Summary of the key results

Kong et al. provides a highly-accessible, "cookbook" format roadmap for generalized protocols for producing soluble HIV Env proteins in forms that largely recapitulate the structural and antigenic properties of native Env proteins - as well as a detailed description of the process, highlighting pitfalls and workarounds. Consideration was duly given to the limitations of general laboratory resources, with an end result of a highly transportable protocol. Discussion of the meaning of the biochemistry and biophysical results in context of the larger goals of the research was exemplary, and greatly appreciated. Therefore, the manuscript as written delivers tremendous value to the community.

B Originality and interest: if not novel, please give references

While building on a wide foundation of previous results, the synthesis of the results presented in this manuscript is quite novel and of direct interest to the community. The extensive presentation and discussion of the results throughout the process also becomes a useful guide for applying rigorous biochemistry in these contexts.

C Data & methodology: validity of approach, quality of data, quality of presentation

Methods and results are clearly presented and thoroughly described. Data validation and quality are completely appropriate and fully support the conclusions as formulated.

D Appropriate use of statistics and treatment of uncertainties

While most of the results are qualitative or semi-quantitative in nature, error estimates are spotty: for example, no estimate of overall coordinate error is provided in Table S1 (very useful at these resolutions), and errors are unevenly reported in Figure S6. Also, "29.3 {plus minus} 0.05 RU" means that the value was truncated one significant digit too early.

E Conclusions: robustness, validity, reliability

The overall impression generated is that the data quite exhaustively support the conclusions and provide considerable internal validation - likely highly reliable.

F Suggested improvements: experiments, data for possible revision

It's not necessary to quote lattice angles for cubic space groups in Table S1.

G References: appropriate credit to previous work?

Literature citations are very complete and thorough.

H Clarity and context: lucidity of abstract/summary, appropriateness of abstract, introduction and conclusions

Extremely well-written; my recommendation is to publish as-is.

Reviewer #2 (Remarks to the Author):

The manuscript by Kong et al discusses a comprehensive, multipronged approach to stabilize soluble trimeric forms of the HIV envelope glycoprotein. It follows reports on the design of stabilized forms of this viral protein by several groups (SOSIP or NFL). The overall goals of the present study is to identify additional modifications that would improve the stability of soluble trimers in their pre-fusion

conformation, they will improve the relative expression of trimers over aggregates and other envelope species (monomers/dimers) during mammalian expression and will improve the antigenic profile of these soluble proteins to better mimic the configuration of the function envelope trimer on the surface of virions (occlusion of non-neutralizing epitope, but exposure of neutralization epitopes). Three major approaches were evaluated: (i) replacing the HR1N domain of gp41 (547-569) by 8 or 10 residues, (ii) replacing the furin cleavage site between gp120 and gp41 (either by a 8 residue linker or by SGS), (iii) replacing the furin cleavage site by longer linkers, and (iv) by a generic HR1 linker.

Each construct was evaluated by complementary methodologies: Expression and purification, thermostability, structural analysis (crystallography and/or EM) and antigenic profile (using broadly neutralizing and non-neutralizing antibodies). To my knowledge, this must be the most complete analysis of diverse soluble HIV envelopes. It is very impressive. Having said that, as it concerns the antigenic structure of these proteins, it does not appear that one redesign approach is clearly superior to the others. Also, regarding the HR1 generic redesign: why is this 'generic' since it benefits only a small fraction of backbones examined? It has no obvious benefit for example to the B/C clade envelope, or the clade B JRFL. Overall the study does not provide a universal way of stabilizing diverse trimers, something that would have been very exciting. Finally, the V3 loop remains well exposed on all these redesigns, as it is exposed on the original SOSIP/NFL designs. The implications of this exposure to the ability of stabilized trimers to elicit broadly neutralizing antibodies are well known to these groups. The above comments and concerns should be addressed by the reviewers:

Minor comments;

1) On the backbone of SOSIP BG505, they replaced the HR1N domain of gp41 (547-569) by 8 or 10 residues, in an effort to stabilize the pre-fusion conformation of the trimeric envelope spike. The two redesigns, appear to reduce exposure of some gp41 non-neutralization epitopes. This is interesting. Pg 12, first paragraph: 'while all three trimers (SOSIP, redesign 1 and redesign 9) showed some V3 exposure.' I would say that based on the nm shifts shown in Fig 3c/d, the V3 is pretty well exposed on these proteins. Even better than the epitopes of VRC01, for example. Therefore, the text should be modified.

2) CST or CSF redesigns: the label on the y axis of Sup Fig 9 need to change to correspond to CST and CSF (rather than CSF and CSF-SOSIP).

Reviewer #3 (Remarks to the Author):

The HIV-1 envelope glycoprotein is a major target for immune recognition, however the development of effective vaccines requires stable immunogens that mimic the native pre-fusion trimer. This is challenging because of the instability of the trimer complex; thus a number of studies have focused on protein modifications to improve antigenic response. Kong et. al, have computationally re-designed the heptad region 1 of Env and replaced the gp120-gp41 cleavage site with different linkers with the goal of improving the yield of trimers that mimic the pre-fusion state. They use gel filtration chromatography, antigenic profiling and x-ray and negative-stain EM structural methods to compare the modified trimers and establish their approach for improving stability. Overall their computational design methods and altered cleavage site appear to enhance the stability of pre-fusion trimers and their structural biology approaches provide validation that the modified trimers are intact. However, these findings overall do not seem significant enough to warrant publication in Nature Communications; a more methods-focused journal seems appropriate for this work. Furthermore, the significance and validation of the 'fusion intermediate' complex is somewhat unclear. I have only minor comments/questions regarding the EM characterization:

1.) Why was cryo-EM not used? It seems for some of the trimer variants and PGV04-bound complexes this study would greatly benefit from higher resolution cryo-EM models. Given current technology and

previous successes with the BG505 SOSIP complex this should be feasible with relatively small amounts of data. Negative-stain EM provides only low-resolution envelopes at $\sim 20\text{\AA}$, which is minimal for the level of structural validation required for these studies.

- 2.) Please provide an example portion of a micrograph image and boxed-out single particles in order to assess the raw data.
- 3.) Please provide information regarding the number of particles that went into each of the reconstructions - how many particles were collected and how many were used to generate the models.
- 4.) How were the half-maps generated for determining the FSC? Was this using the currently accepted 'Gold Standard' method? Also, the resolution value by the FSC method is generally considered an estimate and not absolute. Please state that these are 'estimated' or 'indicated' resolutions.
- 5.) Please indicate in the figure legends what the colors correspond to for the ribbon models that are docked into the EM maps, especially for PGV04. Indicating in the figures what the different lobes of density correspond to would also be helpful as well.
- 6.) How were the two different EM maps for the BG505 CSF complex determined? 3D classification? How these two 'morphologies' were determined is not explained and leads to questions about whether the differences are real or an artifact.
- 7.) Please show the common lines initial model in the supplemental. Was one model used for all reconstructions? If different models were generated for each dataset then the differences in the final models could be due to model bias.
- 8.) The reconstructions for the unliganded complexes in 4D and 5D seem poorly resolved with a lot of empty density that was unable to be docked with the structure. What was the rationale for the density threshold value that was used? A view of the maps at an increased threshold would be useful to see if the areas of higher density are centered with the model.